# Hexokinase-I directly binds to a charged membrane-buried glutamate of mitochondrial VDAC1 and VDAC2
Sebastian Bieker [1,2,6], Michael Timme[1,2,6], Nils Woge[1,2], Dina G. Hassan[1,2,3], Chelsea M. Brown [4], Siewert J. Marrink [4], Manuel N. Melo [5] ✉ & Joost C. M. Holthuis [1,2] ✉

Binding of hexokinase HKI to mitochondrial voltage-dependent anion channels (VDACs) has far-reaching physiological implications. However, the structural basis of this interaction is unclear. Combining computer simulations with experiments in cells, we here show that complex assembly relies on intimate contacts between the *N*-terminal α-helix of HKI and a charged membrane-buried glutamate on the outer wall of VDAC1 and VDAC2. Protonation of this residue blocks complex formation in silico while acidification of the cytosol causes a reversable release of HKI from mitochondria. Membrane insertion of HKI occurs adjacent to the bilayer-facing glutamate where a pair of polar channel residues mediates a marked thinning of the cytosolic leaflet. Disrupting the membrane thinning capacity of VDAC1 dramatically impairs its ability to bind HKI in silico and in cells. Our data reveal key topological and mechanistic insights into HKI-VDAC complex assembly that may benefit the development of therapeutics to counter pathogenic imbalances in this process.

Voltage-dependent anion channels (VDACs) are abundant β-barrel proteins in the outer membrane of mitochondria (OMM) that serve as the main conduits for the large flux of ions, ATP/ADP, NAD + /NADH and Krebs' cycle intermediates from and into mitochondria[1,2]. In mammals, three isoforms exist (VDAC1-3) with non-redundant functions[3,4]. VDAC1 and VDAC2 are the most abundantly expressed isoforms in most tissues. Besides their central role in controlling the flow of metabolites across the OMM, both isoforms act as scramblases that mediate phospholipid import into mitochondria[5]. Additionally, VDAC1 and VDAC2 function as dynamic translocation platforms for a variety of proteins that control the permeability of the OMM for cytochrome *c* to either promote or prevent mitochondrial apoptosis. VDAC binding partners include the pro-apoptotic Bcl-2 proteins BAX and BAK[6–8], which mediate the decisive step in OMM permeabilization by which cytochrome *c* and other apoptogenic factors are released into the cytosol to trigger the apoptotic cascade[9]. Moreover, ceramides, central intermediates of sphingolipid metabolism, exert their pro-apoptotic activity, at least in part, by interacting directly with VDAC2[10].

VDAC1 and VDAC2 also function as the physiological receptors of hexokinases (HKs). These enzymes phosphorylate glucose to generate glucose-6-phosphate (G-6-P), an ATP-dependent reaction that serves as entry point for glucose into the glycolytic pathway for energy production or, alternatively, into the pentose phosphate pathway to generate anabolic intermediates[11]. Elevated levels of mitochondrially bound HK isoforms HKI and HKII lead to a high rate of glycolysis and lactate production, a metabolic signature referred to as the Warburg effect[12]. This metabolic switch from oxidative to glycolytic metabolism is a central hallmark of tumor progression, allowing pre-malignant lesions to maintain a high metabolic rate in oxygen-deprived avascular environments[13–15]. Moreover, mitochondrially bound HKs protect cancer cells from drug-induced mitochondrial apoptosis by diminishing the propensity of VDACs to interact with pro-apoptotic Bcl-2 proteins BAX and BAK[16–18]. Conversely, a reduction in HKI concentration in the spinal cord is thought to enhance binding of VDAC1 to specific amyothrophic lateral sclerosis type I-associated variants of superoxide dismutase 1 (SOD1), thereby promoting formation of toxic SOD1 aggregates, mitochondrial dysfunction and cell death in motor neurons[19,20].

The importance of HKI-VDAC interactions in carcinogenesis and neurodegenerative disease has prompted a search for small molecules and peptides capable of disrupting or stabilizing this protein-protein complex[21–23]. However, these efforts are hampered by a lack of structural

[1]Molecular Cell Biology Division, Department of Biology/Chemistry, University of Osnabrück, 49076 Osnabrück, Germany. [2]Center for Cellular Nanoanalytics, Osnabrück University, Artilleriestraße 77, 49076 Osnabrück, Germany. [3]Department of Environmental Medical Sciences, Faculty of Graduate Studies and Environmental Research, Ain Shams University, Cairo, Egypt. [4]Groningen Biomolecular Sciences and Biotechnology Institute, University of Groningen, Nijenborgh 7, 9747 AG Groningen, The Netherlands. [5]Instituto de Tecnologia Química e Biológica António Xavier, Universidade Nova de Lisboa, Av. da República, 2780-157 Oeiras, Portugal. [6]These authors contributed equally: Sebastian Bieker, Michael Timme. ✉e-mail: m.n.melo@itqb.unl.pt; holthuis@uos.de

insights into how HKI and VDAC assemble into a complex. Like HKII, HKI contains a short *N*-terminal, 20-amino acid hydrophobic α-helix that enables OMM binding, presumably through its interaction with VDACs[24–26]. Two protein-protein docking studies reported models for complex formation based on a direct plugging of the *N*-terminal helices of HKI/HKII into the pore of VDAC1[27,28]. A significant shortcoming of these models is that they fail to address a critical role of a membrane-buried glutamate at position 73 (E73) located on the outside wall of VDAC1 in HKI binding[29,30]. Moreover, a molecular docking simulation study revealed a high-affinity binding site for a peptide mimicking the *N*-terminus of HKI on the outside wall of VDAC1 in close proximity to the bilayer-facing E73 residue[31]. Another modeling study postulated that HKII initially binds the OMM through insertion of its hydrophobic *N*-terminus into the cytosolic leaflet and then interacts with the outer wall of VDAC1 to form a binary complex[32]. Whether the interaction of HKI with VDACs follows a similar scenario remains to be established. At present, the membrane topology or sidedness of VDAC channels has not been definitively assigned, with complementary experimental approaches yielding divergent and contradicting results[33–35]. Knowledge of the actual topography of VDACs is a prerequisite for any comprehensive analysis of their role as mitochondrial scaffolds for a broad variety of proteins.

Here, we combined molecular dynamics simulations with experimental studies in cells to define the structural and topological determinants that govern HKI binding to VDAC1 and VDAC2. We find that complex assembly critically relies on direct interactions between the *N*-terminal α-helix of HKI and a membrane-buried, deprotonated glutamate on the outer wall of both channel isomers. Protonation of this residue abolished complex assembly in simulations. Consistent with this result, we show that VDAC-dependent mitochondrial translocation of a reporter carrying the *N*-terminal α-helix of HKI is exquisitely sensitive to fluctuations in cytosolic pH. Moreover, we find that a pair of polar channel residues flanking the membrane-buried glutamate causes a marked thinning of the cytosolic leaflet, providing a low-energy passageway for HKI to facilitate complex assembly. Taken together, our data offer fundamental mechanistic insights into HKI-VDAC complex formation and indicate that the *C*-termini of VDAC channels must face the intermembrane space to provide functional binding platforms for HKI.

## Results
### A membrane-buried Glu in VDACs is critical for stabilizing the mitochondrial pool of HKI
The bulk of HKI normally resides on mitochondria, with VDACs serving as essential binding platforms. While VDAC1 is widely viewed as principal HKI docking site, the role of VDAC2 is less well defined. As expected, GFP-tagged HKI expressed in HeLa cells extensively co-localized with the OMM marker Tom20 (Fig. 1a; Supplementary Fig. 1a). Removal of either VDAC1 or VDAC2 did not significantly affect mitochondrial localization of HKI-GFP. However, loss of both channels abolished mitochondrial localization of the enzyme and caused its accumulation in the cytosol, even though a portion of the enzyme was found associated with the ER and plasma membrane (Fig. 1a, b; Supplementary Fig. 1a, b; Supplementary Fig. 2a). Moreover, endogenous HKI protein levels were significantly reduced in VDAC1/2 double KO cells while subcellular fractionation experiments showed that in these cells, endogenous HKI primarily resides in the cytosol (Supplementary Fig. 2c, d). Reintroducing VDAC1 or VDAC2 into VDAC1/2 double KO cells restored both mitochondrial localization and expression of HKI (Fig. 1a, b; Supplementary Fig. 2e). These data indicate that VDAC1 and VDAC2 each contribute to stabilizing the mitochondrial pool of HKI.

Both VDAC1 and VDAC2 harbor a uniquely positioned glutamate (Glu) in the transmembrane region of β-strand 4 – Glu73 in VDAC1 and Glu84 in VDAC2 – that faces the bilayer's hydrophobic core. Prior work revealed that Glu73 in VDAC1 is required for HKI binding[30]. Consistent with this, substitution of Gln for Glu73 in VDAC1 abolished its ability to restore mitochondrial localization and expression of HKI in VDAC1/2

double KO cells (Fig. 1a, b; Supplementary Fig. 2e). Likewise, a VDAC2 mutant in which Gln was substituted for Glu84 failed to stabilize the mitochondrial HKI pool. In contrast, substitution of Asp for Glu73 in VDAC1 or Glu84 in VDAC2 yielded a channel that supported mitochondrial recruitment of HKI to a level beyond that observed for its wild type counterpart (Fig. 1a, b). Taken together, these results suggest that HKI binding to VDAC1 and VDAC2 critically relies on a negatively-charged, membrane-buried Glu residue on the outer channel wall.

### Mitochondrial recruitment of HKI is mediated by its *N*-terminal α-helix
HKI contains an *N*-terminal α-helix of 20-amino acids (HKI-*N*) that enables binding to the OMM[24], presumably by interacting directly with VDACs. As expected, a truncated HKI variant lacking this region (HKIΔ2-14) failed to localize to mitochondria and displayed a cytosolic distribution (Fig. 1c). To confirm that HKI-N alone is sufficient for mitochondrial localization, we fused the 17 *N*-terminal amino acids of HKI to a HaLo-Tag and expressed the construct in HeLa cells. In wild-type cells, HKI-N extensively co-localized with OMM marker Tom20. In contrast, when expressed in VDAC1/2 double KO cells, HKI-N failed to target mitochondria and localized to the cytosol (Fig. 1d). This indicates that HKI binds VDACs primarily via its *N*-terminal helix, possibly involving direct contact with the bilayer-facing Glu (Fig. 1e).

The *N*-terminal helix of HKI has been shown to bind membranes even in the absence of VDACs, presumably owing to its partially hydrophobic nature[36]. This implies that mitochondrial recruitment of HKI involves two consecutive steps, namely insertion of its *N*-terminal helix in the cytosolic leaflet of the OMM followed by VDAC binding to form a binary complex. HeliQuest analysis[37] revealed that HKI-N forms an α-helix with an apolar face composed mostly of non-polar and hydrophobic residues and a polar face primarily containing hydrophilic and charged residues (Fig. 2a, b). The amphipathic nature of HKI-N predicts a membrane binding mode whereby its apolar face engages with the hydrophobic membrane core and the polar face with the lipid head groups (Fig. 2c).

To investigate the membrane binding affinity of the HKI-N, we performed coarse-grain molecular dynamics (CG-MD) simulations using the Martini3 forcefield[38,39]. A bilayer mimicking the OMM was constructed[40] and an α-helical peptide comprising HKI-N with an additional Gln at its *C*-terminus (corresponding to Gln18 in HKI) was restrained onto the cytosolic membrane surface. After lifting the restraints, the desorption of the peptide into the aqueous phase was monitored over time[41]. The HKI-N peptide remained membrane-bound, with its apolar face buried into the hydrophobic membrane core and with residence times of >5000 ns. Leu7 is a key component of the membrane-oriented HKI-N apolar face, sitting at its very center (Fig. 2b), and thus likely in constant contact with the hydrophobic membrane core. Substitution of Gln for Leu7 shortened the HKI-N membrane residence time to ~350 ns (Fig. 2d), supporting a critical role of the apolar face in membrane binding. Moreover, substitution of Gln for Leu7 in GFP-HKI abolished its mitochondrial localization in HeLa cells (Fig. 2e, f). Together, these data suggest that membrane insertion of its *N*-terminal α-helix is a prerequisite for HKI binding to VDAC in the OMM.

### HKI-N binding to VDACs is directly controlled by protonation of the membrane-buried Glu
To elucidate the structural basis of HKI-VDAC complex formation, we next performed CG-MD simulations of HKI-N binding to VDAC1 and VDAC2. As the foregoing experiments suggested that HKI-VDAC complex assembly requires a negatively-charged, membrane-buried Glu residue on the outer channel wall (Fig. 1a, b), we first set out to estimate the pKa values of the corresponding Glu residues in VDAC1 and VDAC2 using titratable Martini simulations[42]. This revealed that the pKa value of Glu73 in VDAC1 is shifted compared to a free glutamate in solution[43] but by less than one unit, i.e. from 4.3 to ~4.8 (Fig. 3b, c). For Glu84 of VDAC2, the estimated pKa value is ~5.1 (Supplementary Fig 3a, b). This indicates that at neutral pH, both Glu73 in VDAC1 and Glu84 in VDAC2 are in their deprotonated, negatively charged

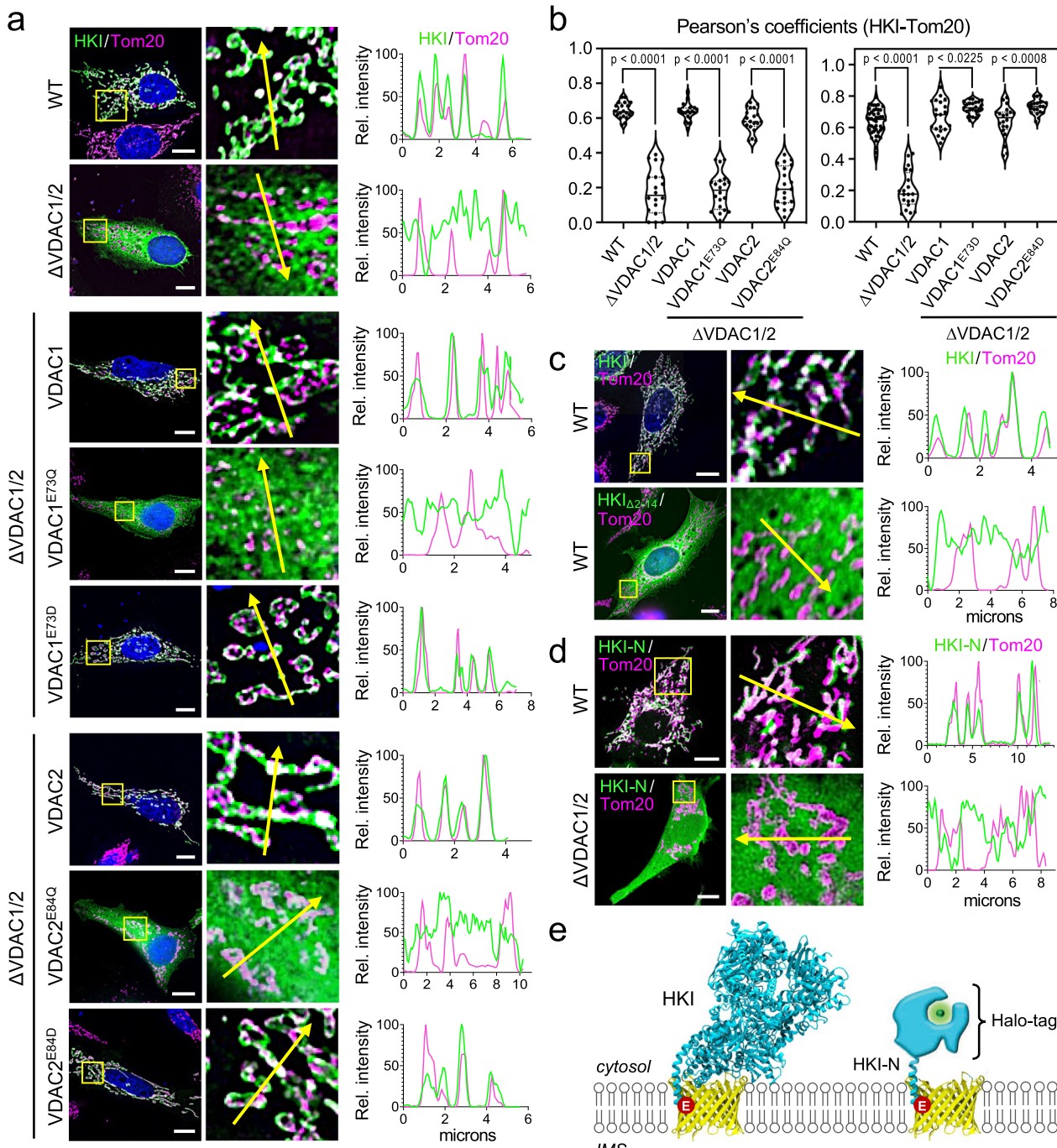

**Fig. 1 | Mitochondrial localization of HKI relies on its *N*-terminal α-helix and a membrane-buried Glu in VDACs. a** Fluorescence images of wild-type (WT) and VDAC1/2-DKO HeLa cells expressing EGFP-tagged HKI (*green*) alone or in combination with HA-tagged VDAC1, VDAC1E73Q, VDAC1E73D, VDAC2, VDAC2E84Q or VDAC2E84D, fixed and then stained with DAPI (*blue*) and an antibody against Tom20 (*magenta*). Line scans showing degree of overlap between HKI and Tom20 signals along the path of the arrow shown in the zoom-in. Scale bar, 10 μm. **b** Pearson's correlation co-efficient analysis between HKI and Tom20 signals in cells as in (a). For each violin plot, the middle line denotes the median, and the top and bottom lines indicate the 75th and 25th percentile. From left to right, *n* = 20 (WT), 20 (VDAC1/2-DKO), 20 (VDAC1/2-DKO + VDAC1), 20 (VDAC1/2-DKO + VDAC1E73Q), 20 (VDAC1/2-DKO + VDAC2), 20 (VDAC1/2-DKO + VDAC2E84Q), 46 (WT), 20 (VDAC1/2-DKO), 20 (VDAC1/2-DKO +

VDAC1), 23 (VDAC1/2-DKO + VDAC1E73D), 20 (VDAC1/2-DKO + VDAC2) and 20 cells (VDAC1/2-DKO + VDAC2E84D) over at least 2 independent experiments. *p* values were calculated by unpaired two-tailed *t* test. **c** Fluorescence images of WT HeLa cells expressing EGFP-tagged HKI or *N*-terminal truncation mutant HKIΔ2-14, fixed and then stained with DAPI (*blue*) and an antibody against Tom20 (*magenta*). Line scans showing degree of overlap between HKI and Tom20 signals along the path of the arrow shown in the zoom-in. Scale bar, 10 μm. **d** Fluorescence images of live WT and VDAC1/2-DKO HeLa cells co-expressing EGFP-tagged Tom20 (*magenta*) and Halo-tagged HKI-N (*N*-terminal HKI residues 1-17, *green*). Line scans showing degree of overlap between HKI-N and Tom20 signals along the path of the arrow shown in the zoom-in. Scale bar, 10 μm. **e** Models of complexes formed between HKI, Halo-tagged HKI-N and VDAC1/2. The membrane-buried Glu is marked in *red*.

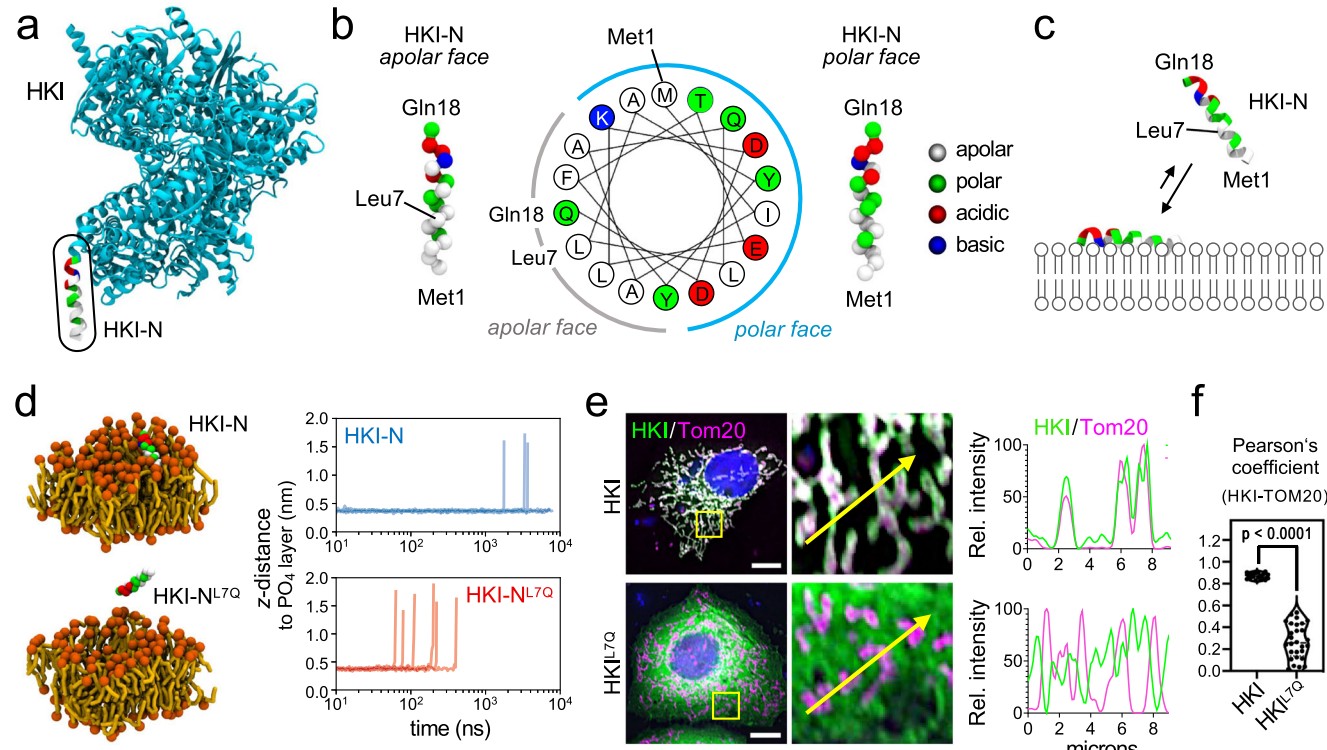

**Fig. 2 | HKI-N binding to membranes. a** Atomic model of HKI (PDB: 1BG3, *cyan*) with the *N*-terminal α-helix (HKI-N) highlighted in residue-type coloring. **b** HeliQuest analysis and VDW/Dynamic Bonds representation of the coarse-grained HKI-N backbone reveals an α-helix with a polar and apolar face. **c** Model predicting that the apolar face of HKI-N mediates membrane binding, with the first half of the α-helix protruding deeper into the membrane bilayer. **d** Membrane residence time analysis of HKI-N and HKI-N$^{L7Q}$ using CG-MD simulations. Helices were bound to OMM-mimicking membranes following the restraining protocol described in Methods. After lifting the restraints, the distance of the helix residue closest to the membrane's top leaflet was measured, until it surpassed 1.4 nm. Stills show representative configurations from each condition. Plots represent the time progression of the helix-membrane distances for six independent replicas per condition, as overlaid semitransparent traces; vertical rises correspond to each trace's membrane-leaving event, from which point that trace is no longer drawn. **e** Fluorescence images of WT HeLa cells expressing EGFP-tagged HKI or HKI$^{L7Q}$ (*green*), fixed and then stained with DAPI (*blue*) and an antibody against Tom20 (*magenta*). Line scans showing degree of overlap between HKI and Tom20 signals along the path of the arrow shown in the zoom-in. Scale bar, 10 μm. **f** Pearson's correlation co-efficient analysis between HKI and Tom20 signals in cells as in (**e**). $n = 20$ (HKI) and 28 cells (HKI$^{L7Q}$) over three independent experiments. *p* values were calculated by unpaired two-tailed *t* test.

state even when residing in the hydrophobic membrane interior. Consequently, we performed CG-MD simulations of HKI-N binding to VDAC1 and VDAC2 with the bilayer-facing Glu in the deprotonated (charged) state. CG-MD-simulations of channels with protonated (neutral) Glu residues served as control to verify the importance of having Glu in its negatively charged form for HKI-N binding. As the membrane topology of VDACs is not known, each channel was also simulated in two orientations, namely with its *C*-terminus facing the cytosol—where HKI-N was present–or the inter-membrane space (IMS). Main simulations were performed in an OMM-mimicking bilayer with an aggregate time of 1.41 ms (Supplementary Table 1)—only attainable using CG-MD.

Strikingly, HKI-N formed stable contacts with both VDAC1 and VDAC2 provided that the channel's *C*-terminus faced the IMS and the membrane-buried Glu was deprotonated (Fig. 3d–f; Supplementary Fig. 3c, d). When these conditions were met, the *N*-terminal half of HKI-N was observed to insert vertically into the cytosolic membrane leaflet along one side of the channel wall and bind directly to Glu73$^-$ in VDAC1 and Glu84$^-$ in VDAC2 (Supplementary Movies 1 and 2). HKI-N residues most frequently in direct contact with the deprotonated Glu were Met1, Ala4 and Gln5, all situated on the same side along the axis of the α-helix (Fig. 3g; Supplementary Fig. 3f), with Met1-Glu73/Glu84 contacts occurring for 17.20 ± 3.11% and 15.68 ± 5.83% of the aggregate simulation time for VDAC1 and VDAC2, respectively (errors indicate SEMs over 3 replicas). These binding events were often observed multiple times per simulation and reached μs durations (Fig. 3e; Supplementary Fig. 3d; Supplementary Fig. 4). Protonation of the membrane-buried Glu severely reduced the contact

prevalence to 0.31 ± 0.25% and 2.14 ± 0.74% for VDAC1 and VDAC2, respectively. Flipping the membrane orientation of the channel in each case completely abolished complex formation. Under these conditions, HKI-N failed to insert into the cytosolic leaflet and no interaction with the bilayer-facing Glu occurred. Instead, contacts with VDAC1 and VDAC2 became random and short-lived (<10 ns), involving channel residues facing the cytosol (Fig. 3f, g; Supplementary Fig. 3e, f).

Consistent with the localization studies of GFP-tagged HKI in HeLa cells (Fig. 1a, b), VDAC channels with a Glu-to-Gln substitution lacked affinity for HKI-N in simulations, regardless of their transbilayer orientation (0.06 ± 0.05% and 1.67 ± 0.31% contact prevalence for VDAC1 and VDAC2, respectively). On the other hand, VDAC channels with a Glu-to-Asp substitution retained the ability to bind HKI-N, provided that the Asp was deprotonated and the channel's *C*-terminus faced the IMS (17.39 ± 4.50% and 36.30 ± 13.26% contact prevalence for VDAC1 and VDAC2, respectively; Supplementary Fig. 5). Collectively, these results indicate that HKI-VDAC binding critically relies on both the membrane topology of VDACs and the protonation state of the bilayer-facing Glu.

### Acidification of cytosolic pH triggers dissociation of HKI-N from mitochondria

To challenge the idea that HKI-VDAC complex formation is controlled by the protonation state of the bilayer-facing Glu, we next investigated the impact of cytosolic acidification on the subcellular distribution of Halo-tagged HKI-N in HeLa cells. Cytosolic pH was adjusted by incubating cells in a buffer with the desired pH in the presence of $H^+/K^+$ ionophore nigericin

(Fig. 4a; Supplementary Fig. 6a). Equilibration of cytosolic pH with the pH of the external buffer was quantitatively assessed with the intracellular pH indicator pHrodo™ Red AM (Supplementary Fig. 6b). To monitor a drop in

cytosolic pH in real time, we took advantage of the fact that the fluorophore of EGFP is more sensitive to acidic pH when compared to mCherry[44] and HaloTag Ligand JF646. Thus, in cells expressing Tom20-EGFP, JF646-

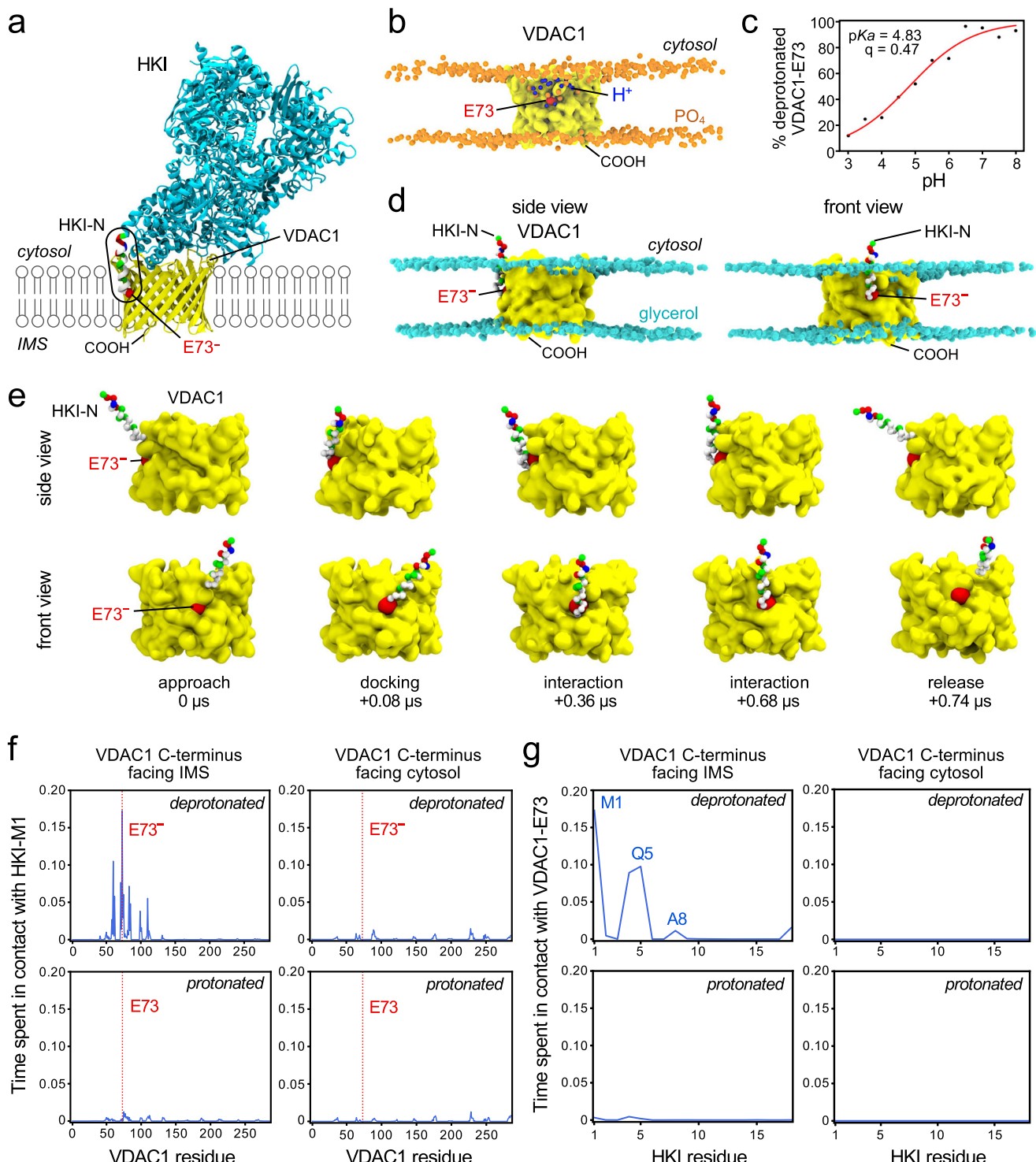

**Fig. 3 | HKI-N binding to VDAC1 is directly controlled by the protonation state of the membrane-buried Glu. a** Atomic model of HKI (*cyan*, with residue-type colored HKI-N) bound to VDAC1 (*yellow*) with the membrane-buried Glu (E73) marked in *red*. **b** Still from a titratable MD simulation of VDAC1 (*yellow*) to evaluate the protonation state of E73 (*red*) at pH 5.0. $PO_4$ groups in the POPC-based bilayer are marked in *orange* and protons are marked in *blue*. **c** Titration curve showing the degree of deprotonation of E73 in VDAC1, simulated at a pH range of 3–8. **d** Stills from an MD simulation showing HKI-N bound to VDAC1 with a deprotonated E73 (*red*) and IMS-facing *C*-terminus. Glycerol groups in the OMM-mimicking bilayer

are marked in *cyan*. **e** Stills from an MD simulation, showing the approach and binding of HKI-N to VDAC1 with a deprotonated E73 (*red*) and IMS-facing *C*-terminus. **f** Relative duration of contacts between HKI-Met1 and specific residues of VDAC1 with a protonated or deprotonated E73 and cytosol- or IMS-facing *C*-terminus simulated in an OMM-mimicking bilayer. Shown are the combined data of three individual replicas with a total simulation time between 169 µs and 211 µs per condition. **g** Relative duration of contacts between VDAC1-E73 and specific residues of HKI-N under the same conditions as in (**f**).

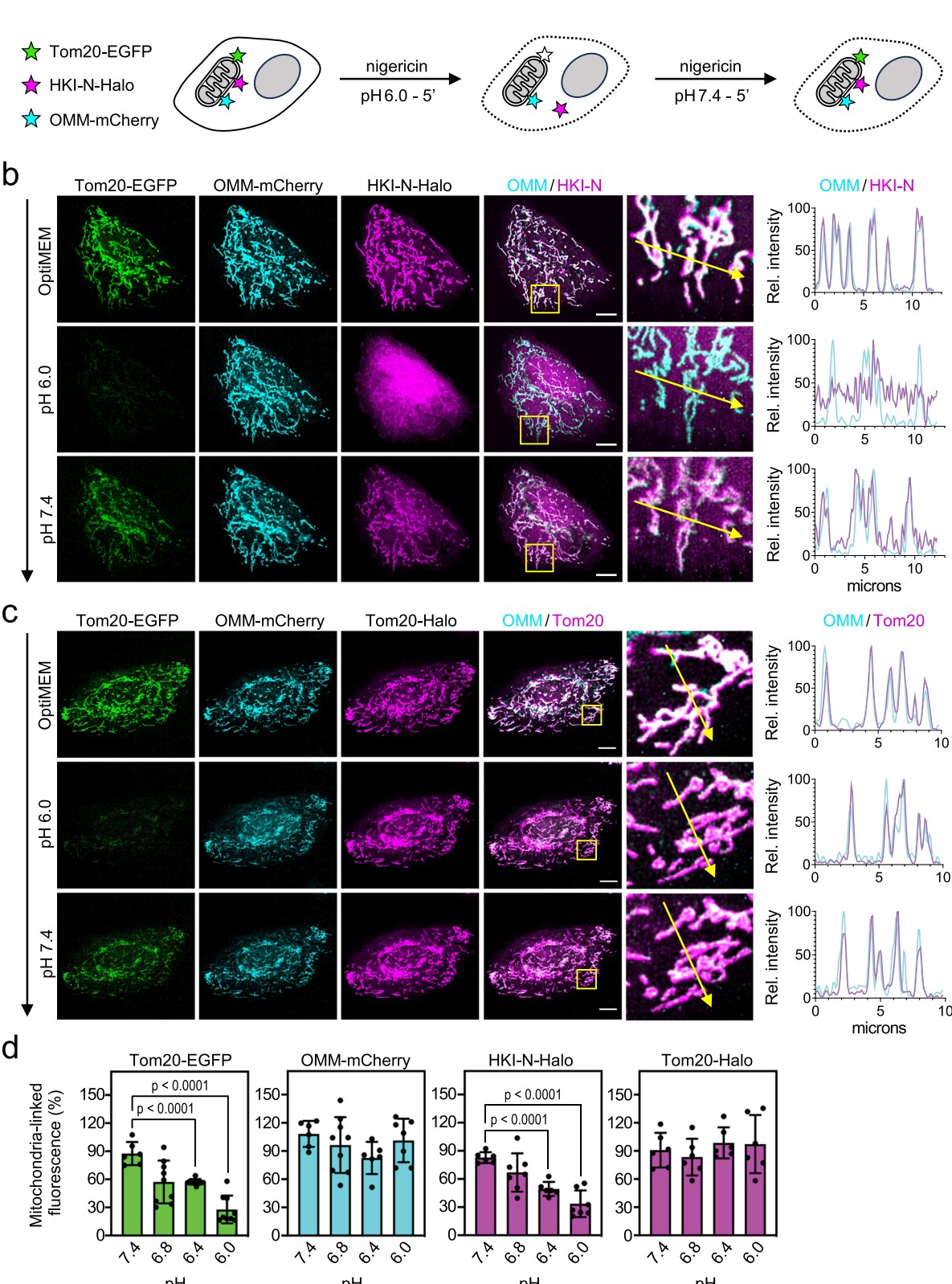

labeled Tom20-Halo and mCherry fused to the OMM anchor of AKAP1 (OMM-mCherry), a shift in cytosolic pH from 7.4 to 6.0 strongly reduced EGFP fluorescence without affecting the other two fluorophores (Fig. 4b, c; Supplementary Fig. 6c, d). Strikingly, acidification of the cytosol readily triggered the translocation of JF646-labeled HKI-N-Halo from

mitochondria into the cytosol. Dissociation of HKI-N-Halo from mitochondria was already measurable when lowering the cytosolic pH to 6.8 and gradually progressed with increased acidification so that at pH 6.0 the bulk of HKI-N-Halo resided in the cytosol (Fig. 4b, d; Supplementary Fig. 6c, d). Raising the cytosolic pH from 6.0 back to 7.4 restored the mitochondrial

**Fig. 4 | Cytosolic pH controls mitochondrial association of HKI-N. a** Schematic outline of experimental strategy to determine the impact of cytosolic acidification on mitochondrial association of HKI-N. **b** Fluorescence images of live HeLa cells co-expressing EGFP-tagged Tom20 (*green*), OMM-mCherry (*cyan*) and Halo-tagged HKI-N (*magenta*) grown in Optimem (top), treated with 10 μM nigericin in pH 6.0 buffer for 5 min (middle) and then with 10 μM nigericin in pH 7.4 buffer for 5 min (bottom). Line scans showing degree of overlap between OMM and HKI-N signals along the path of the arrow shown in the zoom-in. Scale bar, 10 μm. **c** Fluorescence images of live HeLa cells co-expressing EGFP-tagged Tom20 (*green*), OMM-anchored mCherry (*cyan*) and Halo-tagged Tom20 (*magenta*) treated as in (b). Line scans showing degree of overlap between OMM and Tom20-Halo signals along the path of the arrow shown in the zoom-in. Scale bar, 10 μm. **d** Quantitative assessment of mitochondria-associated levels of OMM-mCherry, Tom20-EGFP, Tom20-Halo and HKI-N-Halo in live HeLa cells after treatment with nigericin in buffer at indicated pH for 5 min. Fluorescence values in corresponding pH buffer were set relative to values of same cell in Opti-MEM. Data are means ± SD, *n* = 6 cells per condition over four independent experiments. *p* values were calculated by unpaired two-tailed *t* test.

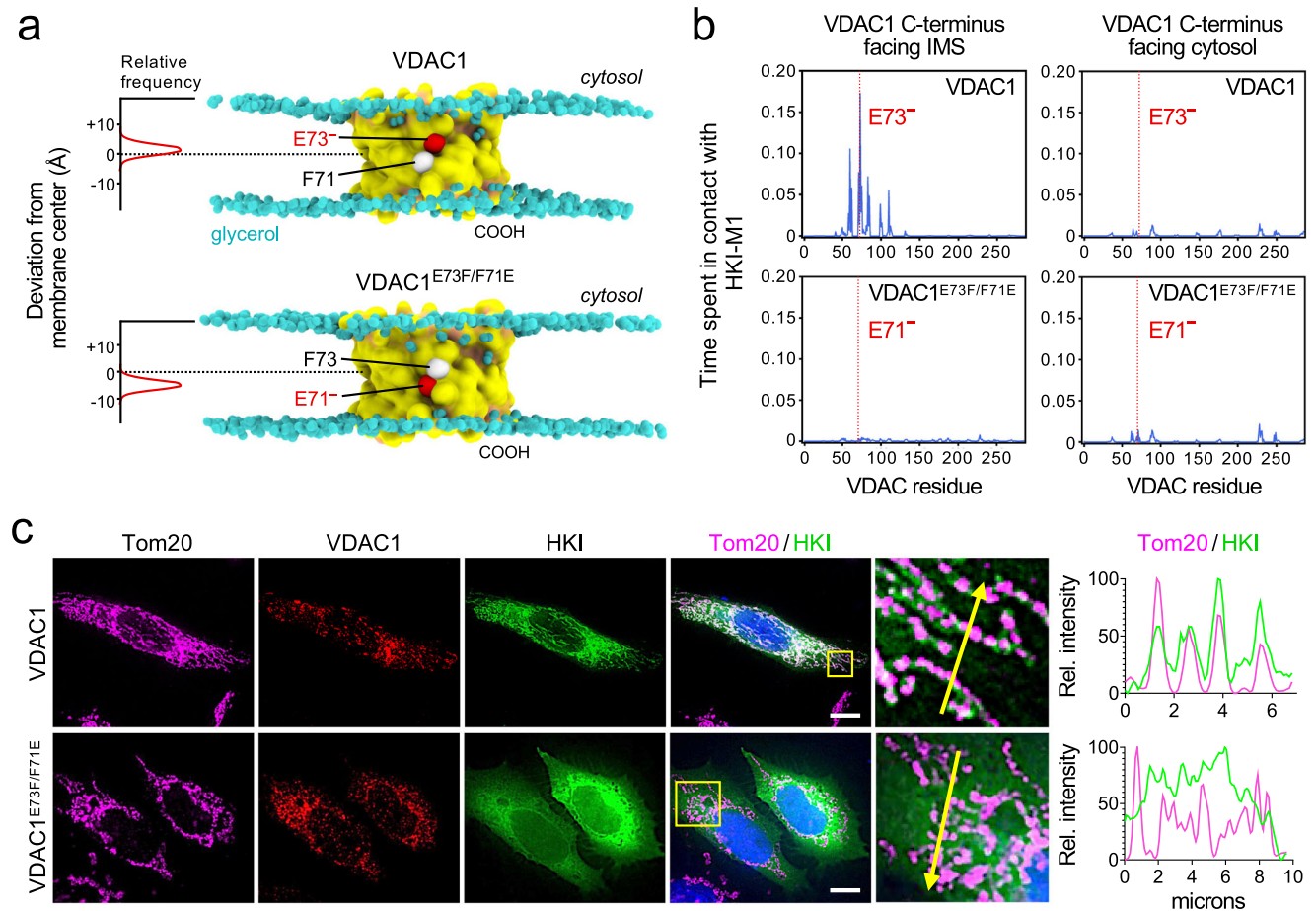

**Fig. 5 | HKI-VDAC binding critically relies on an asymmetric positioning of the membrane-buried Glu. a** Stills from MD simulations of VDAC1 and VDAC1^{E73F/F71E} with the membrane-facing Glu and Phe residues at positions 71 and 73 represented as red and white balls, respectively. The graphs show the position of Glu73 in VDAC1 and Glu71 in VDAC1^{E73F/F71E} relative to the membrane center (dashed line) over the course of a simulation. **b** Relative duration of contacts between HKI-Met1 and specific residues of VDAC1 or VDAC1^{E73F/F71E} with cytosol- or IMS-facing *C*-termini. Data for VDAC1 are taken from Fig. 3f and shown for comparison. For VDAC1^{E73F/F71E} data of three individual simulations were combined with a total simulation time between 148 μs and 162 μs per condition. **c** Fluorescence images of VDAC1/2-DKO HeLa cells co-expressing EGFP-tagged HKI (*green*) and HA-tagged VDAC1 or VDAC1^{E73F/F71E}, fixed and then stained with DAPI (*blue*) and antibodies against the HA-epitope (*red*) and Tom20 (*magenta*). Line scans showing degree of overlap between HKI and Tom20 signals along the path of the arrow shown in the zoom-in. Scale bar, 10 μm.

localization of HKI-N-Halo (Fig. 4b). Consistent with the CG-MD simulations, these results support the notion that HKI-VDAC binding is controlled by the protonation state of the bilayer-facing Glu even though we cannot exclude that protonation of additional acidic residues also play a role.

## HKI-VDAC binding critically relies on an asymmetric positioning of the membrane-buried Glu

The foregoing CG-MD simulations revealed that the transbilayer orientation of VDACs is a critical determinant of HKI binding (Fig. 3f, g; Supplementary Fig. 3e, f). Interestingly, we noticed that the membrane-buried Glu in VDACs is asymmetrically positioned a few Å away from the bilayer center and resides in the cytosolic leaflet when the channel's *C*-terminus faces the IMS, the orientation compatible with HKI binding (Fig. 5a). We

therefore hypothesized that channels with the opposite topology may fail to bind HKI because the membrane-buried Glu in that orientation lies too deep in the lipid bilayer for the enzyme's *N*-terminal α-helix to make stable contacts. To verify this idea, we substituted Phe for Glu73 and Glu for Phe71 in VDAC1, effectively creating a channel in which the asymmetric position of the membrane-buried Glu is flipped across the bilayer center (Fig. 5a). Next, we performed CG-MD simulations to probe HKI-N binding to the VDAC1^{E73F/F71E} mutant channel in both membrane orientations and with a deprotonated Glu. Unlike VDAC1, the VDAC1^{E73F/F71E} variant was unable to form stable contacts with HKI-N irrespective of its transbilayer orientation (Fig. 5b). Moreover, unlike VDAC1, the VDAC1^{E73F/F71E} variant completely failed to restore mitochondrial localization of GFP-HKI in VDAC1/2-double KO cells (Fig. 5c). These results indicate that bilayer depth of the charged Glu on the outer channel wall, although critical, is not the sole

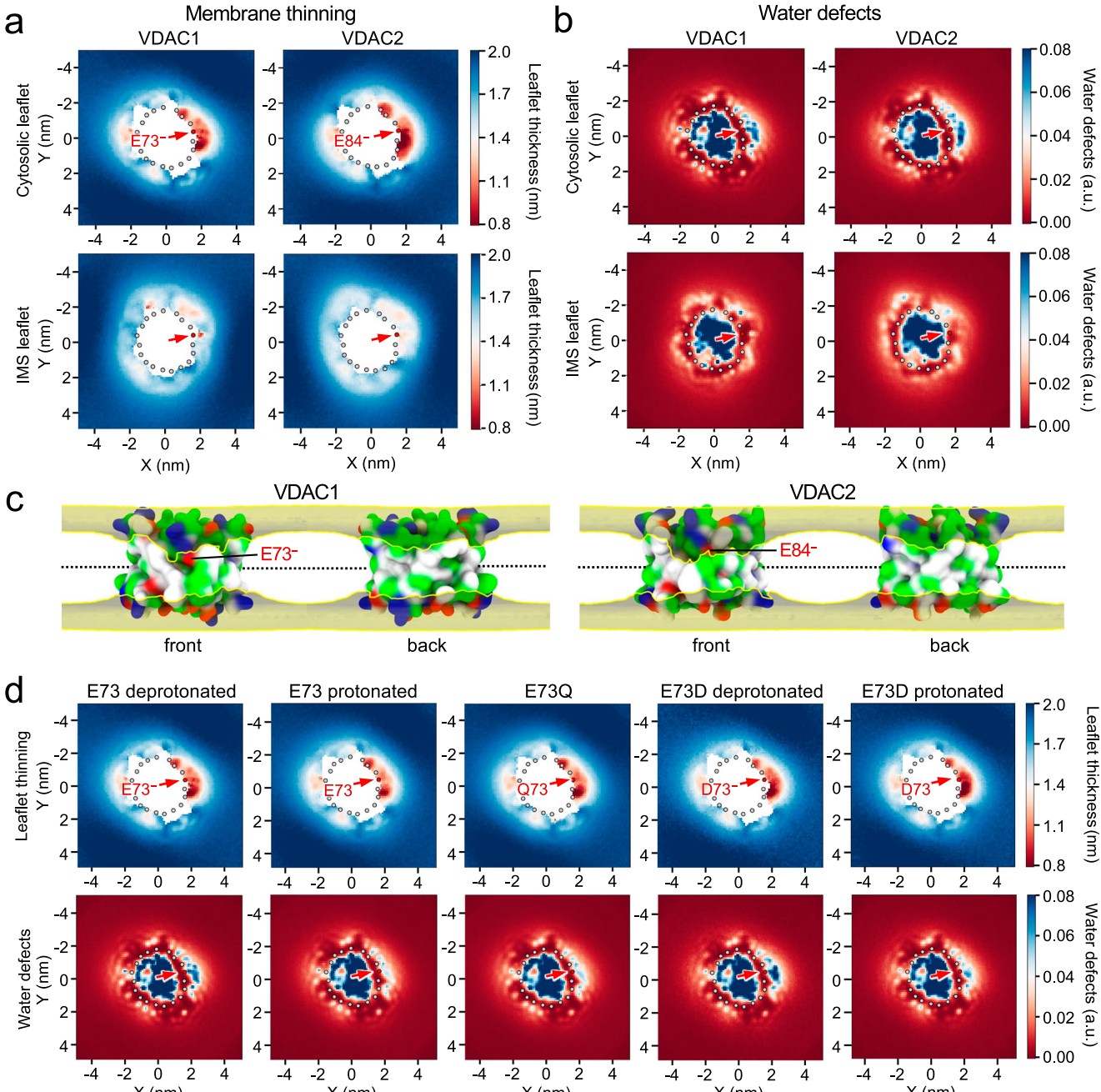

**Fig. 6 | VDAC channels cause lipid packing defects and membrane leaflet thinning proximal to the bilayer-facing Glu.** (**a**) Leaflet-specific membrane thinning graphs of VDAC1 and VDAC2 simulated in a POPC bilayer with *C*-termini facing the IMS leaflet. Gray spheres indicate the VDAC backbone and the position of the bilayer facing Glu is marked by an arrow. Membrane thinning was calculated as the average distance of the lipid backbone phosphates to the global membrane center. (**b**) Leaflet-specific water defect graphs of VDAC1 and VDAC2 simulated as in (a). Water defects were calculated as the amount of water molecules detected within a z-distance of 1.5 nm to the global membrane center. (**c**) Occupancies of lipid $PO_4$ groups in simulations of VDAC1 and VDAC2 as in (a). Occupancy surfaces enclose volumes with average occupancy of 0.5% or greater. The position of the bilayer-facing Glu is marked. (**d**) Cytosolic leaflet thinning and water defect graphs of VDAC1, VDAC1[E73Q] and VDAC1[E73D] simulated in a POPC bilayer with *C*-termini facing the IMS (bottom) leaflet. The bilayer-facing acidic residues were protonated or deprotonated, as indicated. Analysis was done as in (a) and (b).

determinant of HKI binding and that other unique features on the membrane-facing surface of VDACs also play a role.

**VDAC channels cause thinning of the lipid monolayer proximal to the membrane-buried Glu**

VDACs experience a global hydrophobic mismatch with the lipid bilayer in which they are inserted in—having a hydrophobic interface of only ~2.4 nm, which is significantly less than that of biological membranes (~4 nm)[45]. Cumulatively with this overall mismatch, previous MD simulations of

VDAC1 revealed additional membrane thinning and water defects near the outward-facing Glu[5,46]. By extending these studies to VDAC1 in its HKI binding-competent orientation (with the channel's *C*-terminus facing the IMS), we found that this localized membrane thinning is mainly confined to the cytosolic leaflet, adjacent to the negatively charged Glu (E73[-]; Fig. 6a). Here, the cytosolic leaflet reached a minimal thickness of 0.71 ± 0.04 nm, which was considerably thinner than the average thickness near the channel wall outside of this region (1.49 ± 0.01 nm) or in the absence of protein (1.94 ± 0.001 nm; Supplementary Fig. 7a, b). In the defect region, we also

observed a large degree of water penetration (Fig. 6b). Simulations of VDAC2 revealed a similar thinning of the cytosolic leaflet along with water defects near the charged Glu (E73⁻; Fig. 6a, b), with a minimal thickness of 0.52 ± 0.05 nm and an average thickness of 1.46 ± 0.01 nm near the channel wall outside of the defect region (Supplementary Fig. 7a, b). When mapping the occupancy of the lipid phosphates or plotting the thickness of VDAC-surrounding lipids, we observed that the region of membrane thinning did not perfectly overlap with the position of the charged Glu (Fig. 6c; Supplementary Fig. 7b). This suggested that membrane thinning may not rely on a charged Glu but rather on outward-facing polar residues in its vicinity. Indeed, protonation of the bilayer-facing Glu or its substitution by Gln in VDAC1 or VDAC2 greatly diminished the water defects in either case, but had little impact on the membrane thinning capacity of the channels (Fig. 6d; Supplementary Fig. 8). When the bilayer-facing Glu was replaced by a deprotonated Asp (D73⁻), membrane thinning and water defects were retained (Fig. 6d; Supplementary Fig. 8). Hence, while the negatively charged Glu creates conditions that facilitate the penetration of water, it appears that the membrane thinning capacity of VDACs is mediated by other residues on the outer channel wall.

### Polar residues proximal to the membrane-buried Glu provide a gateway for HKI-VDAC binding

We considered that thinning of the cytosolic leaflet near the membrane-buried Glu of VDACs may provide a low-energy passageway for the *N*-terminal α-helix of HKI to facilitate HKI-VDAC binding. A close inspection of the outer wall of VDAC1 in areas exhibiting the highest degree of membrane thinning revealed two polar residues, Thr77 and Ser101, which are positioned within close range of the membrane-buried Glu (Fig. 7a). In CG-MD simulations, substitution of Leu for Thr77 or Ser101 in each case led to a localized but marked reduction in the membrane thinning capacity of VDAC1 carrying a charged Glu (Fig. 7a, b; Supplementary Fig. 7c). The areas occupied by Thr77 and Ser101 each had its own local minimum of leaflet thickness that was selectively abolished by mutation. The thickness on the S101 side displayed the lowest local minimum and corresponds to the global minimum (0.71 ± 0.04 nm). For that reason, the T77L mutation did not affect the global minimum (0.65 ± 0.01 nm). Upon introducing the S101L mutation, the minimum on the T77 side becomes the new global minimum with a higher thickness value (0.90 ± 0.01 nm). When the two mutations were combined, leaflet thinning in the region proximal to the charged Glu was further reduced (1.11 ± 0.01 nm), essentially abolishing the leaflet thinning specific to this region (Fig. 7a, b; Supplementary Fig. 7c). This was accompanied by a substantial reduction in water defects (Fig. 7b). These results indicate that Thr77 and Ser101 each contribute to a local distortion of the cytosolic membrane leaflet, possibly facilitating access of HKI to the charged, membrane-buried Glu. Consistent with this idea, CG-MD simulations revealed that substitution of Leu for Thr77 or Ser101 in VDAC1 diminished contacts between HKI-N and the charged Glu (from 17.20 ± 3.11% to 10.77 ± 4.75% and 11.11 ± 4.08% contact prevalence for T77L and S101L, respectively; Fig. 7c, d). Combining these substitutions further reduced HKI-N binding to a 4.57 ± 2.20% contact prevalence. Looking at contact lifetime distributions, it can be seen that these mutations affect binding by reducing the on-rate rather than off-rate of the binding process (Supplementary Fig. 4), indicating that the polar face surrounding the Glu acts indeed as an access pathway. Importantly, the diminished capacity of the mutant channels to bind HKI-N in silico strongly correlated with an impaired ability of these channels to restore mitochondrial recruitment of HKI in VDAC1/2-double KO cells (Fig. 7e, f). Collectively, these results indicate that Thr77 and Ser101 are core components of a membrane thinning pathway by which the *N*-terminal α-helix of HKI gains access to the membrane-buried Glu of VDACs, thereby providing a gateway for HKI-VDAC binding.

### Discussion

While binding of HKI to mitochondrial VDACs is crucial for cell growth and survival, the structural basis of HKI-VDAC complex assembly is not known.

Using a CG-MD simulations approach complemented with functional studies in cells, we identified core structural and physicochemical features that govern binding of HKI to VDAC1 and VDAC2. As schematically outlined in Fig. 8, our results indicate that a bilayer-facing negatively charged Glu on the outer channel wall plays a crucial role in HKI binding by promoting stable contacts between the channel and the enzyme's amphipathic *N*-terminal α-helix (HKI-N). Protonation of the Glu residue abolishes HKI-N binding in simulations while transient acidification of the cytosol causes a reversable release of HKI-N from mitochondria. Membrane insertion of HKI occurs adjacent to the charged Glu where a pair of polar channel residues causes a marked thinning of the cytosolic membrane leaflet, creating a funnel that likely serves as low-energy passageway for the enzyme's *N*-terminal α-helix to facilitate complex assembly. Consistent with this model, we found that disrupting the membrane thinning capacity of VDAC1 significantly impaired its ability to bind HKI both in silico and in cells.

In line with previous work[47], we demonstrate that HKI-N is essential and sufficient for VDAC binding. However, HKI-N can also bind membranes independently of VDACs[36]. Breaking the apolar face of HKI-N by a single point mutation significantly weakened membrane binding in silico and abolished mitochondrial localization of HKI in VDAC1/2-expressing cells. From this we infer that membrane partitioning of HKI-N is a prerequisite for VDAC binding. Our findings are hard to reconcile with a previous model of HKI-VDAC complex formation that is based on direct plugging of HKI-N into the channel's central pore[27]. Instead, our data indicate that HKI-VDAC complex assembly is a multistep process whereby HKI initially binds the OMM through membrane adsorption involving the apolar interface of HKI-N. We envision that thinning of the cytosolic membrane leaflet by a pair of polar channel residues, Thr77 and Ser101 in VDAC1, creates a funnel that serves as thermodynamic trap for HKI binding by enabling the enzyme's *N*-terminal α-helix to tilt and insert at the VDAC/membrane interface to become aligned for stable interactions with the charged Glu on the outer channel wall.

Additionally, our data provide important clues regarding the transbilayer orientation of VDAC channels in the OMM. The sidedness of these β-barrel proteins has been probed with various approaches without reaching general consensus. For instance, studies on human VDAC1 carrying a cleavage site for cytosolic caspases indicate that the channel's *C*-terminus faces the IMS[34]. In contrast, a split-NeonGreen complementation study suggests that the *C*-terminus of human VDAC2 faces the cytosol[35]. Based on packing analysis of murine VDAC1 crystals in a lipidic environment, Ujwal et al[48]. proposed that VDACs are dual topology membrane proteins that may achieve anti-parallel arrangements in the OMM. However, our MD simulations clearly indicate that HKI-VDAC complex formation is only possible with channels in one orientation, namely whereby their *C*-termini face the IMS. It is only in this orientation that the polar channel residues critical for membrane-thinning are positioned accurately to establish a passageway for cytosolic HKI to reach the bilayer-facing Glu and form a stable complex. While our findings do not rule out the possibility of a dual topology of VDAC channels, they clearly indicate that only one of the two possible transbilayer orientations provides a functional binding platform for HKI.

Titratable MD simulations of VDAC1 and VDAC2 revealed that at neutral pH, the bilayer-facing Glu is predominantly in its deprotonated, fully negatively-charged state. Although it is energetically unfavorable for a charged residue to be exposed to the hydrophobic membrane interior, membrane thinning imposed by polar residues in close proximity of the bilayer-facing Glu may explain why its pKa value is shifted by less than one unit in comparison to a free Glu. Converging lines of evidence indicate that the protonation status of the bilayer-facing Glu is a key determinant of HKI binding. To begin with, protonation of this Glu in VDAC1 and VDAC2 in each case proved sufficient to abrogate HKI-N binding in simulations. Replacing Glu with the non-titratable Gln abolished HKI-N binding to VDAC channels in simulations and disrupted VDAC-dependent mitochondrial localization of HKI in cells. Conversely, replacing Glu for titratable Asp promoted complex formation both in silico and in cells. Mild acidification of the cytosol from pH 7.4 to 6.0 instantly dissociated HKI-N

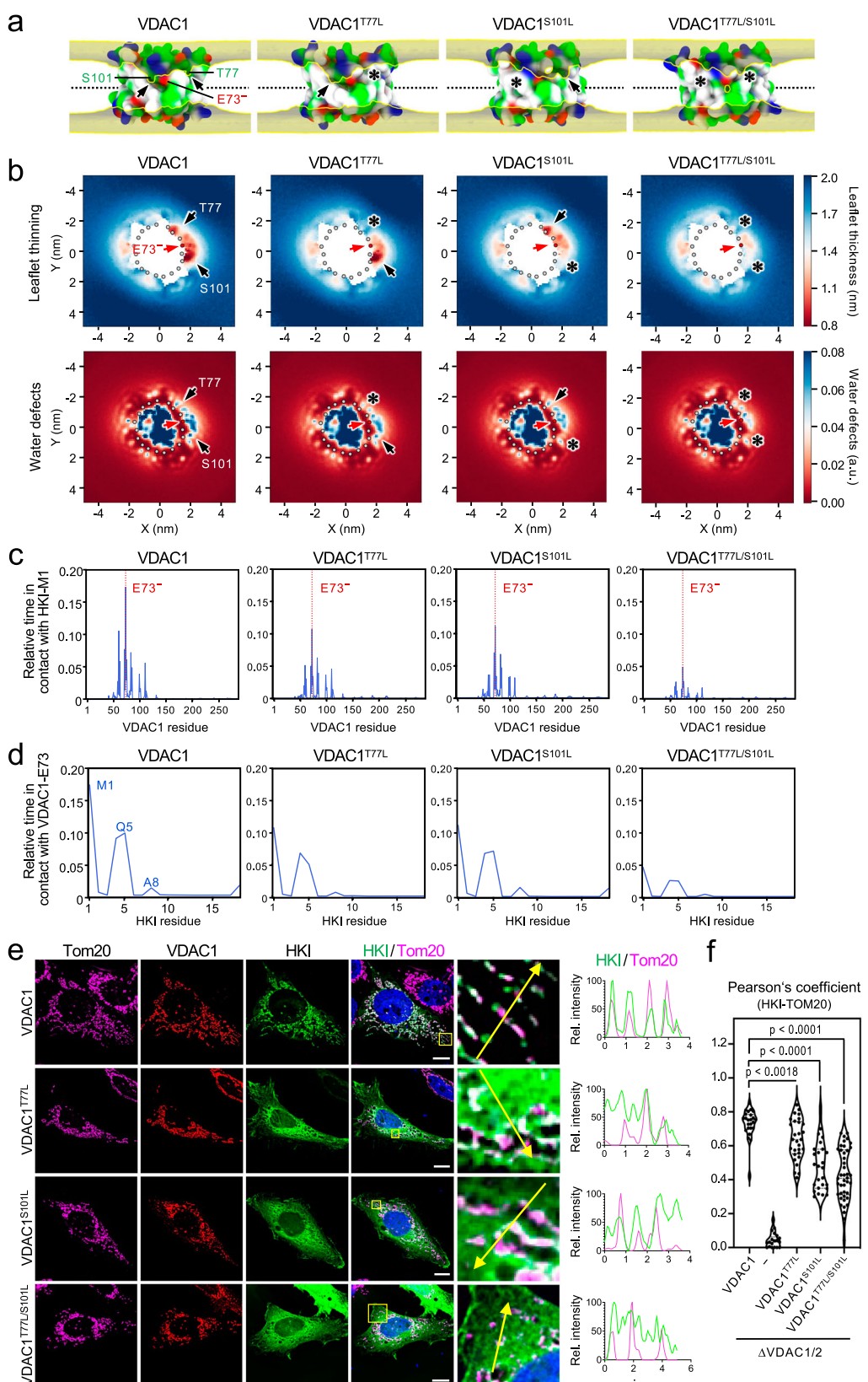

from mitochondria in cells whereas raising the pH back to 7.4 readily restored its mitochondrial localization. While the ability of cellular pH to modulate HKI binding to mitochondria via VDACs has been reported before[49], our present findings indicate that pH-dependent protonation of the bilayer-facing Glu in VDACs plays a direct and decisive role.

A previous study revealed that pH-dependent protonation of the bilayer-facing Glu in VDAC1 promotes formation of a specific VDAC1 dimer with a protein-protein interface that overlaps with the HKI binding site reported in here[50]. Moreover, VDACs reconstituted in OMM-mimicking lipid bilayers adopt various oligomeric structures that

**Fig. 7 | Membrane leaflet thinning and HKI-VDAC1 binding critically rely on channel residues Thr77 and Ser101. a** Occupancy maps of lipid backbone phosphates in simulations of VDAC1, VDAC1[T77L], VDAC1[S101L] and VDAC1[T77L/S101L] in a DOPC bilayer. Positions of polar residues T77 and S101 are indicated. Analysis was done as in Fig. 6d. The VDAC1 panel is from Fig. 6c and shown as reference.
**b** Cytosolic leaflet thinning and water defect graphs of VDAC1, VDAC1[T77L], VDAC1[S101L] and VDAC1[T77L/S101L] simulated in a DOPC bilayer with *C*-termini facing the IMS leaflet. Analysis was done as in Fig. 6a, b. **c** Relative duration of contacts between HKI-Met1 and specific residues of VDAC1, VDAC1[T77L], VDAC1[S101L] and VDAC1[T77L/S101L] simulated in OMM-mimicking bilayers with IMS-facing *C*-termini and a deprotonated E73. Shown are the combined data of three individual replicas with

a total simulation time between 169 μs and 172 μs per condition. (**d**) Relative duration of contacts between VDAC1-E73 and specific residues of HKI-N under the same conditions as in (**c**). (**e**) Fluorescence images of VDAC1/2-DKO HeLa cells co-expressing EGFP-tagged HKI (*green*) and HA-tagged VDAC1, VDAC1[T77L], VDAC1[S101L] and VDAC1[T77L/S101L], fixed and then stained with DAPI (*blue*) and antibodies against Tom20 (*magenta*) and the HA-epitope (*red*). Line scans showing degree of overlap between HKI and Tom20 signals along the path of the arrow shown in the zoom-in. Scale bar, 10 μm. (**f**) Pearson's correlation co-efficient analysis between HKI and Tom20 signals in cells as in (**e**). From left to right, *n* = 23 (VDAC1), 26 (-), 25 (VDAC1[T77L]), 23 (VDAC2[S101L]) and 36 cells (VDAC1[T77L/S101L]) over at least two independent experiments. *p* values were calculated by unpaired two-tailed *t* test.

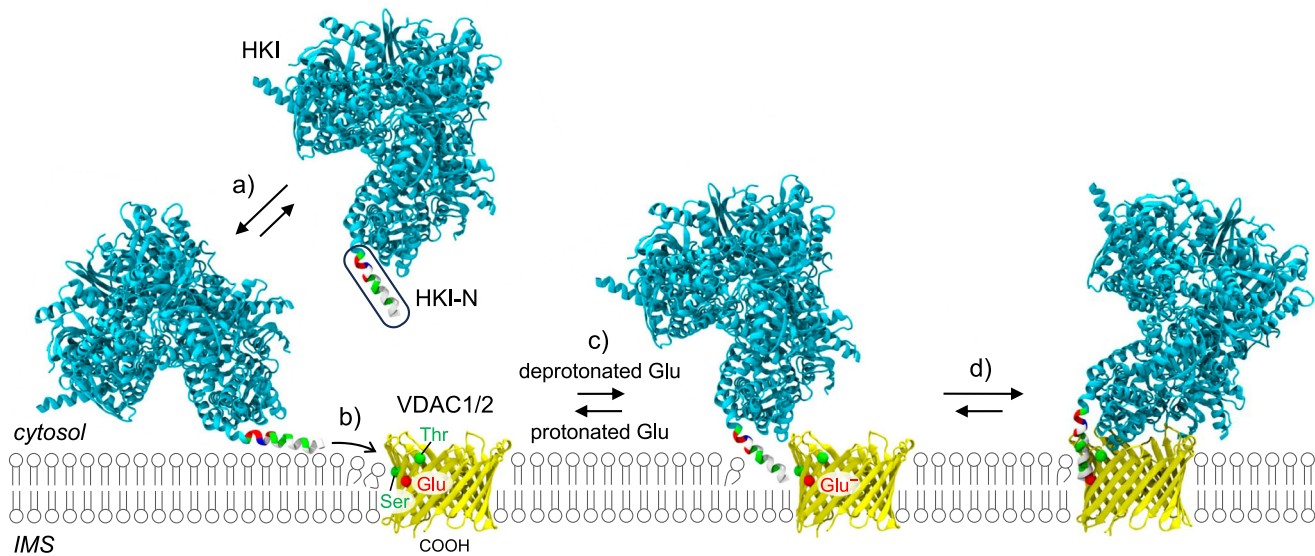

**Fig. 8 | Model of HKI-VDAC complex formation.** HKI-VDAC complex assembly is a multistep process, comprising the following steps: (**a**) HKI binds the OMM through membrane absorption of its *N*-terminal amphipathic helix, HK-N. **b** HK-N inserts into the membrane at a site where a pair of polar channel residues proximal to the bilayer-facing Glu causes a thinning of the cytosolic leaflet. **c** HKI-N undergoes tilting to become aligned for stable interactions with the deprotonated (charged) bilayer-facing Glu on the channel wall. **d** the HKI–VDAC complex is stabilized. See main text for further details.

resemble those observed in the native OMM[45]. While our study describes how protonation of the bilayer-facing Glu influences binary complex formation between VDAC and HKI, the oligomeric organization of VDACs in the OMM adds another level of regulation by which mitochondrial recruitment of HKI can be modulated. There is also evidence that post-translational modifications of HKI influence VDAC binding. Thus, acetylation of Lys15 and Lys21 in HKI was found to promote its VDAC-dependent mitochondrial association whereas deacetylation of these residues by the deacetylase SIRT2 stimulates translocation of the enzyme into the cytosol[11].

Binding of HKI to mitochondrial VDACs has important physiological consequences, from modulating inflammatory responses to promoting cell growth and survival in highly glycolytic tumors. Multiple studies revealed that binding of HKI to VDACs protect tumor cells from permeabilization of the OMM and cytosolic release of cytochrome *c*, an event that marks a point of no return in mitochondrial apoptosis[17,18,51]. Mitochondria-bound HKI confers apoptosis resistance in human colon cancer cells by accelerating retrotranslocation of truncated BID, BAX and BAK from mitochondria[52]. Binding of HKI to mitochondrial VDACs also determines whether the product of the enzyme (G6P) is catabolized through glycolysis or shunted through the anabolic pentose phosphate pathway (PPP). While dissociation of VDAC-HKI complexes shifts the glucose flux to the PPP, leading to increased inflammation and decreased cell survival[11], mild alkalization of cytosolic pH pushes glucose metabolism toward glycolytic flux by augmenting VDAC-HKI binding[49]. Cellular alkalinity is a hallmark of malignancy[53]. Increased intracellular pH is an early event in cancer development[54] and can induce dysplasia in the absence of activated

oncogenes[55]. Its stabilizing effect on VDAC-HKI complexes may facilitate disease progression by promoting glycolysis and apoptosis resistance, thus providing rapidly growing tumor cells with important metabolic and survival benefits. In this context, the oncogenic potential of a somatic missense mutation p.E73D in VDAC1 identified in a colon adenocarcinoma (COSV54738458; cancer.sanger.ac.uk/cosmic) merits further investigation given our present finding that it promotes HKI binding. We previously identified a role of VDAC2 as direct effector in ceramide-induced mitochondrial apoptosis and found that this function critically relies on the channel's charged membrane-buried Glu (E84), which mediates direct contacts with the ceramide head group[10,56]. Our finding that HKI and ceramides share a common binding site on VDACs points at a potential mechanism by which ceramides exert their widely acclaimed tumor suppressor activities[56–58]. Future studies should reveal whether ceramides compete directly with HKI for binding to the charged Glu on the VDAC channel wall and whether their anti-neoplastic activity is linked to a displacement of HKI from mitochondria.

Given the importance of the HKI-VDAC liaison for neoplastic cell growth and survival, disruption of this binary protein complex has been identified as potentially effective therapeutic anti-cancer strategy[21,59]. Additionally, a reduced HKI interaction with VDACs has been recognized as causal factor in demyelinating peripheral neuropathies[60,61]. This has spurred the development of HKI-mimicking peptides as tools for studying the demyelination process and as therapeutics for treating neurodegenerative diseases[20,23,31]. Our present findings provide a molecular framework for the development of novel therapeutic compounds to target pathogenic imbalances in HKI-VDAC complex assembly.

## Methods

### Antibodies

Antibodies used were mouse monoclonal anti-Tom20 (Millipore, MABT166, clone 2F8.1, IF 1:200), mouse monoclonal anti-mitochondrial surface protein p60 (Millipore, MAB1273, IB 1:1,000), rabbit polyclonal anti-HA (Invitrogen, 71-5500; clone SG77, IF 1:100), rabbit polyclonal anti-HKI (Cell Signaling, 2024-s, IB 1:1000), anti-HKII (Cell Signaling, 2867-s, IB 1:1,000), rabbit monoclonal anti-VDAC1 (Cell Signaling, 4661-s, IB 1:1,000), goat polyclonal anti-VDAC2 (Abcam, ab37985, IB 1:4,000) and mouse monoclonal anti-β-actin (Sigma, A1978, IB 1:50,000). HRP-conjugated goat anti-mouse IgG (Thermo Fisher Scientific; 31430). HRP-conjugated goat anti-mouse IgG (31430; IB 1:5,000) and HRP-conjugated donkey anti-goat IgG (PA1-28664; IB 1:5,000) were from Thermo Fisher Scientific. Cy$^{TM}$-dye-conjugated donkey anti-mouse (715-225-150, 715-162-150, and 715-175-150; IF 1:200 each) and donkey anti-rabbit (711-225-152, 711-175-150 and 711-165-150; IF 1:200 each) were from Jackson ImmunoResearch Europe Ltd.

### DNA constructs

pEGFP-HKI encoding rat HKI tagged with EGFP at its C-terminus was described in ref. [62]. pSEMS-HKI-N-Halo encoding the first 17 residues of rat HKI fused to a HaloTag was created by fusion PCR using NEBuilder HiFi DNA assembly kit (New England Biolabs, E5520) and the amplified DNA fragment inserted via EcoRI and XhoI sites into expression vector pSEMS-Halo (Covalys Biosciences). Expression constructs pSEMS-OMM(Akap1)-mCherry, pSEMS-Tom20-Halo and pSEMS-Tom20-EGFP were described in refs. [63,64]. Human VDAC1 and VDAC2 carrying a C-terminal HA tag (YPYDVPDYA) were PCR amplified from corresponding cDNAs using Phusion high-fidelity DNA polymerase (Thermo Fischer Scientific) and inserted via NheI and XbaI sites into mammalian expression vector pcDNA3.1 (+). For retroviral transduction of cells, DNA fragments encoding HA-tagged VDAC1 and VDAC2 were created by PCR and inserted via NotI and XhoI sites into lentiviral expression vector pLNCX2 (Takara Bio, USA). Single amino acid substitutions were introduced using NEB's site-directed mutagenesis kit (New England Biolabs, E0552S). Primers used for cloning and site-directed mutagenesis are listed in Supplementary Table 2. All expression constructs were verified by DNA sequencing.

### Cell culture and transfection

Human cervical carcinoma HeLa cells (ATCC CCL-2) were cultured in Dulbecco's modified Eagle's medium (DMEM, PAN-Biotech, P04-04510) supplemented with 4.5 g/l glucose, 2 mM L-glutamine and 10% FBS. Human colon carcinoma HCT116 cells (ATCC CCL-247) were cultured in McCoy's medium supplemented with 10% FBS. Human embryonic kidney HEK293T cells (ATCC CRL-3216) were cultured in DMEM supplemented with 10% FBS. All cell lines were free of mycoplasma contaminations as determined routinely by DAPI staining or PCR assay. Cells were transfected with DNA constructs using polyethylenimine (PEI, Polysciences, Inc., 24765-100). In brief, 3 µg of DNA was dissolved in 200 µl of 150 mM NaCl, mixed with PEI reagent (2 µl/µg DNA), incubated for 15 min at RT, and then added dropwise to cells seeded in a well of a 6-well plate (Sarstedt AG & Co. KG, 83.3920). After 4 h of incubation, cells were washed with PBS, cultured overnight and then processed for fluorescence microscopy.

### Generation of HKI-KO and VDAC-DKO cell lines

To knock out HKI in HeLa cells, we obtained a mix of three different CRISPR/Cas9 plasmids for the corresponding gene from Santa Cruz Biotechnology (sc-401753-KO-2). The HKI-specific gRNA sequences were: A/sense, 5'-CAGAGCTTACCGATTCTCGC-3'; B/sense, 5'-AGATGTTG CCAACATTCGTA-3'; C/sense, 5'-GCAGATCTGCCAGCGAGAAT-3'. HeLa cells were transfected with the plasmid mix and after 24 h, single GFP-expressing cells were sorted via fluorescence activated cell sorting (FACS, SH800S, Sony) and grown in 96-well plates. Individual clones were expanded and analyzed for HKI expression by immunoblot analysis. To

knock out VDAC1 and VDAC2 in HeLa cells, we obtained a mix of three different CRISPR/Cas9 plasmids per gene and the corresponding HDR plasmids from Santa Cruz (sc-418200, sc-416966). The VDAC1-specific gRNA sequences were: A/sense, 5'-TTGAAGGAATTTACAAGCTC-3'; B/sense, 5'-CGAATCCATGTCGCAGCCC-3'; C/sense, 5'-CTTACAC ATTAGTGTGAAGC-3'. The VDAC2-specific gRNA sequences were: A/sense, 5'-AGAAATCGCAATTGAAGACC-3'; B/sense, 5'-GCCCTTAA GCAGCACAGCAT-3'; C/sense, 5'-TAATGTGACTCTCAAGTCCT-3'. HeLa cells were transfected with both plamid mixes and grown for 48 h without selection. Next, cells were grown for 2 weeks under selective pressure with 2 µg/ml puromycin. Individual drug-resistant clones were picked and analyzed for VDAC1 and VDAC2 expression by immunoblot analysis. A VDAC1/2 double KO cell line was generated from ΔVDAC1 cells as described above following ejection of the puromycin selectable marker using Cre vector (Santa Cruz, sc-418923) according to the manufacturer's instructions. HCT116 VDAC1-KO, VDAC2-KO and VDAC1/2-double KO cells were previously described[10].

### Retroviral transduction

VDAC1/2 double KO cells stably expressing HA-tagged VDAC1, VDAC1$^{E73Q}$, VDAC2 or VDAC2$^{E84Q}$ were created by retroviral transduction. To this end, HEK293T cells were co-transfected with pLNX2-VDAC-HA expression constructs and packaging vectors (Clontech) using Lipofectamine 3000 (Invitrogen) according to manufacturer's instructions. The culture medium was changed 6 h post transfection. After 48 h, the retrovirus-containing medium was harvested, filtered through a 0.45 µm filter, mixed 1:1 (v/v) with McCoy's growth medium, supplemented with 8 µg/ml polybrene and used to transduce the VDAC1/2 double KO cells. Hygromycin (300 µg/ml) was added 6 h post-infection and selective medium was exchanged daily. After 3-5 days, positively transduced cells were selected and analyzed for expression of HA-tagged VDACs by immunoblot analysis and immunofluorescence microscopy using an anti-HA antibody.

### Subcellular fractionation

HCT116 cells were grown to 75% confluency, washed twice with ice-cold 0.25 M sucrose and scraped using a rubber policeman in IM medium (250 mM Mannitol, 5 mM HEPES, 0.5 mM EGTA, pH 7.4) supplemented with protease inhibitor cocktail (1 ug/ml apoprotein, 1 ug/ml leupeptin, 1 ug/ml pepstatin, 5 ug/ml antipain, 157 ug/ml benzamidine) and 0.1 mM PMSF. Cells were homogenized on ice by flushing 20-30 times through a Balch Homogenizer with a 8.008 mm diameter tungsten-carbide ball. Nuclei and cellular debris were removed by centrifugation at 600 gmax at 4℃ for 5 min. The resultant post-nuclear supernatant was centrifuged at 10,300 gmax at 4℃ for 10 min to collect mitochondria. The mitochondrial pellet was resuspended in an ice-cold IM buffer, washed twice in the same buffer and resuspended in 5 volumes R buffer (0.25 M sucrose, 10 mM Tris-HCl, pH 7.4) supplemented with protease inhibitor cocktail and 0.1 mM PMSF. The supernatant from the 10,300 g spin was centrifuged at 100,000 gmax at 4℃ for 1 h to separate the cytosol from microsomes. The mitochondrial and cytosolic fractions were subjected to immunoblot analysis using antibodies against HKI, HKII, VDAC1, VDAC2 and mitochondrial marker p60.

### Manipulation of cytosolic pH

pH adjustment buffer was prepared as in Zaki et al[65]. and contained 2 mM CaCl$_2$, 5 mM KCl, 138 mM NaCl, 1 mM MgCl$_2$, 10 mM D-glucose and 10 mM HEPES (pH7.4, 6.8) or 10 mM MES (pH6.4, 6.0) and supplemented with 100 µg/ml penicillin-streptomycin (PAN-Biotech, P06-07100). The pH of the buffer was adjusted with 0.1 M NaOH or HCl just before use. Nigericin (Enzo Life Sciences, BML-CA421-0005) was dissolved in DMSO to obtain a 10 mM stock solution and added to the pH adjustment buffer at a final concentration of 10 µM. HeLa HKI KO cells seeded in 8-well slides and transfected with Tom20-EGFP, OMM-mCherry, HKI-N-Halo or Tom20-Halo were labeled with Janelia Fluor® HaloTag® Ligand JF646 (Promega, CS315110) at a final concentration of 30 nM for 30 min. Cells possessing a

well-developed mitochondrial network were selected and imaged in Opti-MEM™ (Gibco, cat#.11058-021) as described below. Next, Opti-MEM™ was replaced with nigericin-containing pH adjustment buffer set at the desired pH and imaging continued after 5 min of incubation. This was repeated after each exchange with nigericin-containing pH adjustment buffer set at a different pH. Equilibration of the cytosolic pH with the pH of externally added pH adjustment buffer was verified using pHrodo™ Red AM intracellular pH indicator dye (Thermo Fisher Scientific, FP35372) according to instructions of the manufacturer.

### Live cell imaging

Live cells were imaged using a Zeiss Cell Observer microscope equipped with a CSU-X1 spinning disk unit (Yokogawa) at 37°C. Images were acquired at magnification of 75.6 x using an Alpha Plan-Apochromat 63x oil immersion objective (NA 1.46) and immersion oil for 37°C (Immersol 518 F, 1.518, Zeiss). Fluorophores used were: EGFP (λex = 488 nm, λem = 509 nm), mCherry (λex = 587 nm, λem = 610 nm), and JF646 (λex = 653 nm, λem = 668 nm). For each cell, a z-stack containing 21 slices of 0.2 μm thickness each was recorded. Images were deconvoluted and corrected for chromatic aberration using Huygens Remote Manager (Scientific Volume Imaging, Netherlands). For chromatic aberration correction, the x,y,z shifts were measured using 100 nm ⌀ multicolor beads and the same microscope settings as used for sample imaging. Calculated shifts (D) in μm for the channels in reference to the GFP channel are listed in Supplementary Table 3. Image processing was performed on the sharpest z-layer using Fiji software (National Institutes of Health, USA, 1.54i). The total fluorescence intensity of the mitochondrial network was quantified using Trainable Weka Segmentation Plugin for ImageJ[66]. Segmentation models of mitochondria, cytosol, and extracellular background were trained uniquely for each cell. After applying a trained model, an intermodes threshold was set on the probability map (Image => Adjust => Threshold) and a binary mask was created. The mask was transformed into a ROI which was added to the original image to measure the mean pixel intensity inside the segmented area. Total pixel intensity at mitochondria was calculated by multiplying mean pixel intensity x total segmented area. Fluorescence values in corresponding pH buffer were set relative to values of same cell in Opti-MEM™. At least 6 different cells per condition were quantified from 4 independent experiments. The contrasts of the images were set in reference to the intensity level of the Opti-MEM™ images of each channel, except for the image of the cell expressing HKI-N-Halo incubated with pH 7.4 buffer in Fig. 4b, where the intensity was increased to improve visualization.

### Immunofluorescence microscopy

HeLa cells grown and transfected on 12 mm sterilized glass coverslips were fixed in 4% (w/v) paraformaldehyde in PBS for 10 min at 37°C. After quenching in 50 mM ammonium chloride, cells were permeabilized using PBS containing 0.1% (w/v) saponin and 0.2% (w/v) BSA, immunostained for Tom20 and HA-tagged VDACs, counterstained with DAPI and mounted on glass slides using ProLong™ Antifade Gold Mountant (Thermo Fisher Scientific, P36934). Cells were imaged using a DeltaVision Elite microscope (GE Healthcare) using a PLAPON 60x oil immersion objective (NA 1.42) and Immoil FC30CC immersion oil (Olympus Life Science, n = 1.518, 23°C). Fluorophores used were: DAPI (λex = 390 nm, λem = 435 nm), Cy2 (λex = 475 nm, λem = 523 nm), Cy3 (λex = 575 nm, λem = 632 nm), and Cy5 (λex = 632 nm, λem = 676 nm). For each cell, a z-stack containing 16 slices of 0.2 μm thickness was recorded. Images were deconvoluted using SoftWoRx 5.5 software and further processed using Fiji software.

### Line scan and Pearson's correlation coefficient analysis

Line scan analysis was done on either Huygens-processed data for live-cell imaging or on deconvoluted immunofluorescence images. For this, 32-bit gray live-cell or 16-bit immunofluorescence images were used. An arrow was drawn through the region of interest (ROI) and the pixel intensity data in this region was derived using Analyze => Plot Profile => List. The data

was normalized using the formula ((Single value-MIN(Values))/(MAX(Values)-MIN(Values)))*100. Pearson's correlation coefficients were calculated using the Costes' automatic threshold. Pearson's values for the experiment shown in Fig. 1b were determined using the Fiji Plugin Coloc 2. For all other experiments Pearson's values were determined using the Image J Macro described in Supplementary Information.

### Computational models and structures

For all-atom models, a structure of rat HKI (PDB: 1BG3 at 2.80 Å resolution)[67] and a refined solution NMR structure of human VDAC1 (PDB: 6TIQ)[68] were used. The same VDAC1 structure was used for all CG-MD simulations. As there is no available structure for human VDAC2, we mutated the structure of human VDAC1 to the sequence of human VDAC2 (UniProtKB: P45880) using the PyMOL software. The VDAC1 structure shares a β-barrel backbone RMSD of 2.03 Å and a mean of 2.09 Å from all 20 structures of the NMR stack with a zebrafish VDAC2 structure (PDB: 4BUM)[69], supporting our assumption of identical secondary structures between VDAC1 and VDAC2. To achieve similar sequence length, we ignored the first eleven residues of VDAC2 and mutated the twelfth residue onto the first residue of VDAC1, thus effectively truncating the N-terminus of VDAC2 by eleven residues. Since the published structure of human HKI (PDB: 1HKB)[70] lacks the N-terminal helix, we used the first 18 residues of the N-terminal helix of rat HKI structure (PDB: 1BG3)[67] for CG-MD simulations as this sequence is identical to that of human HKI and is a complete helical segment (the structure in 1BG3 kinks immediately after Gln18). Another reason to use the first 18 residues of HKI (compared to the 17 residue peptide used in experiments with cells) is that including Gln18 might better represent a possible anchoring of the C- versus the N-terminus to the membrane interface, and preemptively satisfy peptide-water interactions. This measure was a precaution to counter the tendency of short Martini 3 peptide helices towards excessive hydrophilicity in membrane contexts[71,72]. All proteins were coarse-grained using the martinize2 script[73]. The HKI-N C-Terminus was protonated to mimic a continuation of the protein. For MD-simulations with HKI-N, VDAC1 was embedded into an OMM-mimicking lipid bilayer with a total of 749 lipids using the insane script[74]. The membrane composition was based on Horvath and Daum[40], with a mixture of POPC/POPE/SAPI/cholesterol (45/33.5/5/16.5, mol%) in the cytosolic leaflet and POPC/POPE/SAPI/cholesterol (52.5/14/19/14.5, mol%) in the IMS-leaflet. To achieve a similar tilt and starting position between the two VDAC isoforms, the VDAC2 backbone particles were then aligned onto the membrane-embedded VDAC1 backbone using the MDAnalysis[41] and MDreader (https://github.com/mnmelo/MDreader) Python packages. In both VDACs, a barrel lumen-facing aspartate (D100 in VDAC1 and D111 in VDAC2) is present, at a position just under the Martini electrostatic interaction cutoff relative to the HKI-N terminal particles when bound to the membrane facing Glu. Two nearby lysine side chains, that would presumably shield the aspartate's anionic charge, fall just outside this electrostatic distance cutoff. Therefore, to avoid over-representing the influence of this aspartate's charge on the HKI-N bound state, it was always considered to be protonated. For membrane thickness, lipid phosphate occupancy and water defects analysis, VDAC1 was embedded into a 100 mol% POPC membrane containing 386 lipids using the insane-script. VDAC2 was aligned onto it as described above. Approximately 150 mM NaCl was added to all systems, with an excess of Na$^+$ to reach charge neutrality.

### MD simulation settings

Six types of systems were simulated, over at least three replicates each (summarized in Supplementary Table 1), for a total simulation time of 3.78 ms: VDAC1 and VDAC2, and their protonated/mutated forms, together with HKI-N in an OMM lipid setting; VDAC1 and VDAC2, and their protonated/mutated forms, without HKI-N in a 100% POPC setting; HKI-N only in an OMM lipid setting; and a protein-free, 100% POPC membrane. The Martini 3 force field[38] with updated lipid models[75] was used for all simulations and the secondary structure of all proteins was restricted

using an elastic-network approach[76], placing harmonic bonds of 500 kJ mol⁻¹ nm⁻² between backbone particles that lie within 0.9 nm after coarse-graining of the reference structures. All simulations were run with GRO-MACS versions 2021[77]. We employed standard Martini 3 parameters to calculate interparticle interactions. For electrostatic interactions, reaction field electrostatics with a Coulombic potential cutoff of 1.1 nm were applied. The relative dielectric constant was set to 15, with an infinite dielectric constant of the reaction field. Van der Waals interactions were modeled by the Lennard-Jones potentials up to a cutoff of 1.1 nm. The particle neighbor list was updated using the Verlet list scheme. All simulations were run at 20 fs time steps. All systems were minimized using a steepest descent algorithm. The systems were then equilibrated for 2 ns at 300 K and 1 bar in the isothermal–isobaric (NpT) ensemble. During equilibration, temperature was controlled with the v-rescale thermostat with a coupling constant of 1.0 ps. Pressure was coupled semiisotropically using the Berendsen barostat, with a coupling constant of 3.0 ps and a compressibility of $4.5 \times 10^{-5}$ bar⁻¹. During the equilibration, HKI-N was adsorbed onto the membrane bilayer using the restraining protocol described in the next subsection. Production runs followed largely the same setup, but with the more formally correct Parrinello-Rahman barostat, used with a coupling constant of 12.0 ps and a compressibility of $3.4 \times 10^{-5}$ bar⁻¹; particle coordinates were saved as a trajectory every 500 ps. Except for the comparison of residence time on the membrane between HKI-N and HKI-N$^{L7Q}$, all other HKI-N simulations were run with a potential on the helix that imposed an effective barrier to crossing the system's periodicity over the z-axis, should HKI-N become desorbed during the simulation. This repulsive harmonic potential of 1000 kJ mol⁻¹ nm⁻² was only imposed within a vertical thickness of 1.5 nm of the z = 0 position, and thus did not influence the membrane-adsorbed state. To analyze the residence time on the membrane between HKI-N and HKI-N$^{L7Q}$, we first equilibrated the adsorption of HKI-N or HKI-N$^{L7Q}$ onto an OMM-mimicking lipid bilayer without VDACs, using the restraint approach described below, and then lifted the restraint for the production run.

### Restrained equilibration of peptide adsorption

To ensure sampling of representative membrane bound configurations of the HKI-N peptide, we employed an equilibration of peptide adsorption using restraints, as described and used elsewhere[41,78]. It should be noted that as a short Martini helix HKI-N has very limited backbone flexibility, and can be thought of as rodlike. In brief, HKI-N was subject to two restraints: i) a flat-bottom harmonic restraint in z (of 500 kJ/mol/nm² force constant), applied to the first and last backbone beads of the peptide and with onset at 3 nm from the bilayer center; it ensured the peptide is in close proximity to the bilayer during equilibration without affecting movement in z towards the membrane center; and ii) a harmonic restraint in x and y (also of 500 kJ/mol/nm² force constant) pinning the backbone bead of a central residue (Tyr9) to the x,y coordinates of the box origin (and thus as far as possible from VDAC, when it is present, since it is initially placed at the box center); this restraint allows the peptide to spin about both the z and the helix's axes, but not to diffuse in the x,y plane. This set of restraints facilitates that the peptide adopts the most favorable orientation and depth when adsorbed onto the membrane, without prematurely interacting with VDAC in less representative configurations. Both restraints are lifted after equilibration to allow free diffusion and interactions of HKI-N. We kept VDACs unrestrained, as the equilibration time of 2 ns was short enough to prevent notable VDAC-diffusion or any contacts with HKI-N.

### Titratable Martini simulations of VDAC channels

To perform the titration of VDAC1 and VDAC2, the corresponding channel structures (see above) were first simulated in a purely POPC membrane using the insane tool[74] and a solvated box with neutralizing ions for 1 μs using a time-step of 20 fs. The final frame then provided the starting structures for the titration simulations. To perform the titration simulations, the protocol described by Martini Sour[79] was used as in ref. 42. Briefly, the side chain of interest (E73 for VDAC1 and E84 for VDAC2) was replaced

with a titratable particle (type P2_4.8) which has the ability to bind a proton particle. The standard Martini water was replaced with titratable Martini water. For each protein, three independent sets of titrations were performed. Each pH (in the range 3-8 with half pH steps) was simulated for 20 ns. Prior to the production simulation, minimization and equilibration steps were performed (2 ns each) at each pH value. The production simulations were calculated using the NPT ensemble, with the temperature set at 298 K and pressure set at 1 bar. To maintain these, a velocity rescale thermostat (time constant 1.0 ps) and the isotropic Parrinello Rahman barostat (time constant 3 ps) were used. All titrations were performed using the stochastic dynamics integrator[80] with a timestep of 10 fs. The PME algorithm was used to calculate electrostatic interactions, with a cutoff value of 1.1 nm. For analysis, only the last 10 ns of each simulation were considered. The scripts to perform the titrations and analysis can be found at https://github.com/fgrunewald/titratable_martini_tools and http://cgmartini.nl. The degree of deprotonation is calculated based on the number of proton particles bound to the titratable site at each pH.

### Simulation analysis

The minimum distance between protein and membrane particles was calculated for each frame of the trajectory and followed/plotted until it exceeded 1.4 nm, upon which desorption was considered to have occurred. HKI-N–VDAC interactions were analyzed as contacts between the proteins' particles within a 0.6 nm cutoff, grouped into contacts per residues (residues were considered in contact if they have any particles in contact). These were plotted as either HKI-N Met1 contacts to any VDAC residues or any HKI-N residue contacting VDAC1/VDAC2 Glu73/84 or the corresponding mutants. In case of VDAC1$^{E73F/F71E}$, contacts with Glu71 were plotted. Contact intensity is shown as the fraction of the total simulation time for which the contact was established. To define contact events and measure their lifetimes, we set Met1-Glu distance cutoffs at 0.6 and 1.0 nm; a contact event was defined as a continuous range of frames in which the distance was always inside the 1.0 nm cutoff, as long as it reached below the 0.6 nm cutoff at some point in that interval. The use of a double cutoff strategy preserves longer contact events by suppressing the frequent spurious binding/unbinding frames that are otherwise counted when a single cutoff is used and the distance fluctuates in its vicinity. Contact durations were histogrammed into logarithmic bins, and plotted after weighting for the total contact duration per bin and normalizing for total simulated time. Average contact lifetimes were also calculated weighting for contact duration. The membrane-depth of the membrane-buried Glu was calculated by first defining the membrane center as the average position of the centers of geometry from each leaflet's lipid backbone glycerols. The z-position of the Glu was then calculated relative to the membrane center over the course of the simulation. To analyze leaflet specific membrane thinning, the absolute value of the difference between the average z-position of the lipid backbone phosphate in a particular leaflet in 0.1 nm bins along the x and y plane and the global membrane center of mass was calculated. Minimum and average membrane thickness values were only considered if the corresponding bin had more than 30 individually measured phosphate counts. To measure membrane thinning around VDACs, the lowest leaflet value in a circle around the protein was determined and plotted. Water defects were calculated with the same bin dimensions as for the thickness analysis. Here, the average number of water molecules inside a cylinder over the course of a simulation was calculated. The cylinder was placed in the geometric center of the protein in the xy-plane with a radius of 2.5 nm and in the geometric center of the membrane in the z-axis with a total height of 3 nm, thus protruding 1.5 nm into each leaflet. The average leaflet thickness of a 100 mol% POPC membrane was approx. 1.95 nm. We analyzed the lipid backbone phosphate occupancy using the VMD VolMap Tool, by gridding the simulation box at a 2 Å spacing and then calculating the presence of a phosphate in each cell relative to the total simulation time, leading to an occupancy range per cell from 0% (never present) to 100% (always present). For the representation, an isooccupancy surface threshold of 0.5% was chosen. To analyze membrane thinning, water defects and phosphate

occupancy, VDACs were centered in the xy-plane and rotationally fit, around the z-axis only, to a common reference.

### Statistics and reproducibility

For microscopy data, sample sizes and number of biological replicates are given in the figure legends. *p* values were calculated by unpaired two-tailed *t* test. For simulations data, reported value uncertainties correspond to standard errors of the mean (SEM) over three replicates, except for contact lifetimes, for which uncertainties correspond to weighted SEMs over several contact events of all replicates.

### Reporting summary

Further information on research design is available in the Nature Portfolio Reporting Summary linked to this article.

### Data availability

All data generated or analyzed in this study are included in the manuscript and supporting files. Source data with sample sizes, number of technical and/or biological replicates, means, standard deviations, and calculated *p* values (where applicable) are provided in the Supplementary Data file. Uncropped scans of immunoblots are provided in Supplementary Information.

### Code availability

Custom code for image analysis is provided in Supplementary Information. Models and code related to MD simulation preparation and analysis have been deposited in the Zenodo repository under doi:10.5281/zenodo.14144611[81].

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

## Acknowledgements
The authors gratefully acknowledge Ladislav Bartos and Robert Vácha (National Centre for Biomolecular Research, Masaryk University, Brno, Czech Republic) for providing the scripts for membrane thinning and water defects analysis, and Varda Shoshan-Barmatz (Ben-Gurion University of the Negev, Israel) for the pEGFP-HKI construct. This work was supported by the Deutsche Forschungsgemeinschaft (378148610 and 448344643 to J.C.M.H.), the German Egyptian Research Long-term Scholarship Program (GERLI project 57222240 to D.G.H.), the European Research Council (ERC Advanced grant 101053661 „COMP-O-CELL" to S.J.M.) and the FCT—Fundação para a Ciência e a Tecnologia I.P. (through MOSTMICRO-ITQB R&D Unit with projects UIDB/04612/2020 and UIDP/04612/2020, and LS4FUTURE Associated Laboratory with projects LA/P/0087/2020 and CEECIND/04124/2017/CP1428/CT0008 to M.N.M.).

## Author contributions
M.N.M. and J.C.M.H. designed the research with critical input from S.B. and M.T.; S.B. performed experiments in cells with critical input from D.H.; M.T. carried out the CG-MD simulations with critical input from N.W.; C.M.B. carried out all titratable MD simulations; J.C.M.H. provided expertise for experiments in cells and helped interpret the data; M.N.M. and S.J.M. provided expertise for CG-MD simulations and helped interpret the data; J.C.M.H. wrote the manuscript; all authors discussed results and commented on the manuscript.

## Funding

## Competing interests
The authors declare no competing interests.
