## [Transparent Peer Review file · Communications Biology]

Hexokinase-I directly binds to a charged membrane-buried glutamate of mitochondrial VDAC1 and VDAC2

Corresponding Author: Professor Joost Holthuis

Version 0:

Reviewer comments:

Reviewer #1

(Remarks to the Author)

The authors are exploring the binding site (s) or interaction site(s) between HK-1 and VDAC1 and 2 using cellular experiments and MD simulation methods. They demonstrated well the interaction between them and described the possible scenario involving membrane topology and negative charge amino acid residues. Further, they showed that protonation plays a significant role in their interaction. In my opinion it is a significant and good piece of work. The authors have done extensive work to investigate the problem, and it deserves appreciation. However, I have a few queries which are listed below.

1. How do the authors justify the use of rat HK-1 in their interaction studies with human VDAC-1/2? Since they are from different species, their function and mode of interaction might be completely different from their observation in comparison to real physiological phenomena. Will it not be better to use HK-1 and VDAC-1/2 from the same species?
2. P9, lines 326, 327: Binding of HKI to VDAC and its dissociation is a normal process leading to switching on & off the glycolytic cycle. And this is known to take place at the neutral pH. However, the authors indicate that HKI-VDAC dissociation takes place on acidification in vitro. How does the dissociation occur in a cell at a neutral pH?
3. P9, lines 343-345: Is alkalinity a cause or effect in malignancy? How does stabilization of VDAC-HKI complex give rise to rapid growth of tumor cells? Is there any time frame beyond which the aforesaid complex is tumor causing? A brief explanation highlighting the mechanism involved for tumor growth in this case is desirable.

Reviewer #2

(Remarks to the Author)

The manuscript by Bieker et al. investigates the specific residue(s) of VDAC1 and VDAC2 involved in the interaction with Hexokinases. This interaction is relevant to mitochondrial physiology and to pathological mechanisms involved in cancer and neurodegeneration, and merits an in-depth study. Briefly, the manuscript focuses on the role of glutamate 73 of VDAC1 and the corresponding glutamate 84 of VDAC2, and how their protonation state drives the interaction with the N-terminal region of Hexokinase 1. The results are largely based on co-localization experiments by confocal microscopy in knock-out HeLa cells, and are supported by molecular dynamics simulations. Overall, the manuscript is well written, the rationale is clear, the experiments and data analysis were performed correctly. Furthermore, the results mostly support the authors' conclusions. However, I have some concerns that should be addressed before publication, as detailed below.

- 1) A previous study by the group of Abramson (PMID: 29279396) has highlighted the involvement of E73 residue of VDAC1 in the protein oligomerization process, a mechanism that largely depends on the protonation state of this residue. However, the authors did not cite the work in the Introduction nor comment on it in the Discussion, and this represents a serious flaw. In my opinion, the authors should contextualize their findings in the light of the previous result.
- 2) The authors have briefly introduced the role of VDAC1 in ALS in lines 64-66, but this sentence is too generic and inaccurate. They, indeed, should be clearly explain that an interaction of VDAC1 with mutated forms of SOD1 was found exclusively in a specific form of the inherited pathology (ALS type 1) and has been demonstrated for specific SOD1 mutations (i.e. G93A, G85R). In fact, mutations in SOD1 gene account for about 20% of the genetic ALS cases that, in turn, represent just the 10% of the overall cases. Authors should also use the full name for SOD1 and not just the acronym.
- 3) In the sentences in line 74-76, the authors should take into account that the involvement of VDAC1 E73 residue in interaction with HK1 has been already proposed by Magri and coauthors. In the paper, already cited by the authors (Ref. no. 54), molecular dynamics simulations were carried out by using a small peptide mimicking the N-terminus of HK1.

4) The sentence in line 109, "Combinatorial loss of VDAC1 and VDAC2 wiped out the mitochondria-associated pool of endogenous HKI", referring to Fig. S2C, is not quite correct. Indeed, from Fig. S2D clearly emerges that the absence of VDAC1 and VDAC2 per se strongly affect the expression of endogenous HKs. Therefore, the subcellular fractionation must be contextualized with this result (i.e., the reduced mitochondrial pool of HKs is principally the result of a reduced expression of HKs in comparison to the other experimental conditions).

5) To performing molecular dynamics simulations, authors added an additional Glu residue to the HK1 N-terminal sequence of 17 residues long amino acid peptide, previously used in the experiments. What is the rationale for this change?

6) Regarding the generation of the 3D model of VDAC2, the authors assumed that the structures of VDAC1 and VDAC2 are identical (as reported in Methods, line 529), despite several differences in the amino acid composition. Based on this consideration, they mutated the structure of VDAC1 by including the additional 11-amino acid portion of VDAC2 at the N-terminal domain. This, however, appears to me a questionable approach since a predicted 3D structure, such as that generated by the popular software AlphaFold, is available.

7) The accession number reported for VDAC2 (line 526) associates with a protein long 310 residues, whereas is known that the main VDAC2 isoform is composed of 294 residues. Please correct.

8) The authors should clearly explain the rationale behind the use of Leu7Gln mutation in CG-MD experiment (lines 152-154) to explain the changes of residence time along the membrane. This approach, in my opinion, is not immediately clear.

9) It is supposed that the pH acidification should not affect the co-localization of blue and magenta signals. However, in Fig. 4C, at pH 6 the colocalization is not appreciable, as instead it is in the Fig. S5D.

Minor comments:

1) In line 43, another reference should be added (PMID: 22020053).

2) The explanation of Fig. 2C in line 142 is not clear. From the figure, it looks like that the apolar face takes contact with the polar head of phospholipids.

3) In line 74, please correct "significant" with "significant".

Reviewer #3

(Remarks to the Author)

The presented work aims to shed light into the mechanism of binding of hexokinase KHI to the mitochondrial voltage-dependent anion channels VDAC1/2. To achieve this goal the authors combine both computational simulations and experimental results. The authors provide evidence for several of the steps likely to be relevant for the binding, such as the importance of the presence of a bilayer-facing negatively charged Glutamic acid present in both VDAC1 and VDAC2, the importance of the N-terminal alpha helix of HKI and the orientation that these channels are most likely to take to allow the binding.

The authors provide an exhaustive and well-designed analysis of the different questions at hand combining coarse grained MD analysis with experimental validation for each step. However, the manuscript could benefit from more clear explanations of the steps followed and analysis performed, in order to improve interpretability and reproducibility of the presented results. My comments would be structured as follows: minor format corrections such as typos and consistency, minor corrections and optional suggestions to improve the manuscript.

Format corrections:

The way that mutations are written in the text is confusing. Usually, one writes first the amino acid that is presented in the wildtype followed by the mutated residue. The authors write those abbreviations correctly when using the one letter codes of the amino acids (for example VDAC1 E73F/F71E in line 226) but they write it the other way around when writing it in an extended way. For example, in line 223-224 "we substituted Phe for Glu73 and Glu for Phe71" should be "we substituted Glu73 for Phe and Phe71 for Glu". This is true for all the mutations described in the manuscript.

There is a typo in the methods section at line 567, seiisotropically should be semiisotropically.

There is no citation for HeliQuest on the manuscript.

The references for each PDB structure should be added next to the PDB codes.

Minor corrections:

The authors performed a quite exhaustive set of analyses. However, in some sections it is difficult to follow what the authors did or how they arrived to those conclusions. Additionally, a more quantitative analysis of the results would help even if the final conclusion is qualitative. The following comments aim to improve on those aspects:

The initial placement and restraining protocol of HKI-N for the adsorption MD studies should be explained in more detail even if it was already described in citation 38. Similarly to how the authors summarize the Titrable Martini simulations setup described in ref 39. The initial placement and restrain setup can heavily influence the final results so it merits a more detail explanation.

On line 152, where the authors study the importance of the apolar face of HKI-N in membrane binding, why was the residue Leu7 selected? All other mutations studied in the manuscript have a detailed explanation of why they were picked (previous knowledge, interactions observed ...). However, this mutation has no apparent reason for being selected. Were other positions studied? Is there any specific analysis performed to select this position? The authors should explain this in more detail.

For the desorption analysis, one should be careful when using only one replicate and using total residence times. It would be also beneficial if the authors added to the text the increments between the helix and membrane distance which were observed several times on the L7Q mutant before the complete desorption criteria was met (figure 2d).

On the paragraph starting at line 176, the authors should add numeric results (fractions of time) extracted from the MD simulations when talking about the frequency or stability of the contacts. It would also be beneficial to add a measure of uncertainty for example by doing the same analysis over different chunks of the obtained trajectories.

On the "VDAC channels cause thinning of the lipid monolayer proximal to the membrane-buried Glu" section, more numerical results from figure 6 should also be added on the text. For example, the authors comment that they observed a membrane thinning reaching about 0.8 nm. In here it would be helpful to provide the membrane size in the conditions where thinning was not observed.

For the membrane thinning analysis, was the whole simulation used or were some frames discarded as equilibration? How uncertain are the estimated averages? What are the observed average membrane sizes over different chunks of the trajectory? This kind of information would support the reliability of the results.

For the "Polar residues proximal to the membrane-buried Glu provide a gateway for HKI-VDAC binding" section, on line 257 I would suggest adding that the described results come from simulations of the mutated Thr77 and Ser101 systems.

Similar to my previous points, for the frequency, contacts and membrane thinning, the authors should add some numeric results to the text to facilitate comparison and provide statistical uncertainty of the obtained results. The uncertainty point is even more important in this section, where the effects observed are less drastic than the ones described on the other sections.

Optional comments:

It would be beneficial for the reproducibility and transparency if the authors could share the input files (structures, scripts to set up the simulations and GROMACS input files such as topology files, mdp files and restrain specification files used on this work).

Although in the supplementary information there is a list of all the simulations performed with their box sizes and simulation lengths, in my opinion it would be beneficial if in the methods section there is a summarized description of all the systems simulated. This would help illustrate all the work performed for the manuscript.

Version 1:

Reviewer comments:

Reviewer #1

(Remarks to the Author)

The authors have responded to my queries 1 & 2 satisfactorily. Regarding the query no. 3 the authors refused to include a discussion on the mechanism involved for tumor growth in this case. They have stated "the issues raised goes beyond the scope of our present study". In my opinion the authors should at least mention this in the revised manuscript.

Reviewer #2

(Remarks to the Author)

The authors have fulfilled all my concerns. In my opinion, the work is now suitable for publication.

Reviewer #3

(Remarks to the Author)

I believe that the authors have addressed all my previous comments and corrections successfully and that the manuscript is ready for publication in its current form.

Response to reviewers' comments

Reviewer #1

The authors are exploring the binding site (s) or interaction site(s) between HK-1 and VDAC1 and 2 using cellular experiments and MD simulation methods. They demonstrated well the interaction between them and described the possible scenario involving membrane topology and negative charge amino acid residues. Further, they showed that protonation plays a significant role in their interaction. In my opinion it is a significant and good piece of work. The authors have done extensive work to investigate the problem, and it deserves appreciation. However, I have a few queries which are listed below.

1) How do the authors justify the use of rat HK-1 in their interaction studies with human VDAC-1/2? Since they are from different species, their function and mode of interaction might be completely different from their observation in comparison to real physiological phenomena. Will it not be better to use HK-1 and VDAC-1/2 from the same species?

The reviewer makes a fair point. Yet, our study primarily focuses on characterizing the minimal structural determinants of HKI-VDAC binding, which for HKI corresponds to the enzyme's N-terminal alpha-helix (residues 1-17). In this region, rat HKI (UniProt entry P05708) and human HKI (UniProt entry P19367) share 100% sequence identity.

2) p9, lines 326, 327: Binding of HKI to VDAC and its dissociation is a normal process leading to switching on & off the glycolytic cycle. And this is known to take place at the neutral pH. However, the authors indicate that HKI-VDAC dissociation takes place on acidification in vitro. How does the dissociation occur in a cell at a neutral pH?

To experimentally validate our molecular dynamics simulations data, we analysed the impact of cytosolic acidification on the VDAC-dependent mitochondrial binding of a HKI mini-reporter, HKI-N, which contains only the first 16 N-terminal residues of the enzyme. While our results indicate that the protonation status of the bilayer-facing Glu in VDACs is a critical determinant of HKI binding, mitochondrial association of HKI is also controlled by other mechanisms. As now outlined in the Discussion (p. 10, lines 365-369), acetylation of Lys15 and Lys21 in HKI was found to promote its mitochondrial association whereas deacetylation of these residues by the deacetylase SIRT2 stimulates translocation of the enzyme into the cytosol (De Jesus et al. 2022; Ref. 11).

3) p9, lines 343-345: Is alkalinity a cause or effect in malignancy? How does stabilization of VDAC-HKI complex give rise to rapid growth of tumor cells? Is there any time frame beyond which the aforesaid complex is tumor causing? A brief explanation highlighting the mechanism involved for tumor growth in this case is desirable.

We appreciate the comments of the reviewer. However, we also feel that elaborating on the issues raised goes beyond the scope of our present study, which is primarily focused on the structural basis of VDAC-HKI complex assembly.

Reviewer #2

The manuscript by Bieker et al. investigates the specific residue(s) of VDAC1 and VDAC2 involved in the interaction with Hexokinases. This interaction is relevant to mitochondrial physiology and to pathological mechanisms involved in cancer and neurodegeneration, and merits an in-depth study. Briefly, the manuscript focuses on the role of glutamate 73 of VDAC1 and the corresponding glutamate 84 of VDAC2, and how their protonation state drives the interaction with the N-terminal region of Hexokinase 1. The results are largely based on co-localization experiments by confocal

microscopy in knock-out HeLa cells, and are supported by molecular dynamics simulations. Overall, the manuscript is well written, the rationale is clear, the experiments and data analysis were performed correctly. Furthermore, the results mostly support the authors' conclusions. However, I have some concerns that should be addressed before publication, as detailed below.

1) A previous study by the group of Abramson (PMID: 29279396) has highlighted the involvement of E73 residue of VDAC1 in the protein oligomerization process, a mechanism that largely depends on the protonation state of this residue. However, the authors did not cite the work in the Introduction nor comment on it in the Discussion, and this represents a serious flaw. In my opinion, the authors should contextualize their findings in the light of the previous result.

We thank the reviewer for pointing out the study by Abramson and colleagues. We have now discussed our findings in the light of both this previous work (Bergdoll et al., 2017 - Ref. 50) and a recent study by Lafargue et al. (bioRxiv 2024.06.26.597124 - Ref. 45), highlighting oligomeric organization of VDACs as additional mechanism for controlling mitochondrial recruitment of HKI (p. 10, lines 359-365).

2) The authors have briefly introduced the role of VDAC1 in ALS in lines 64-66, but this sentence is too generic and inaccurate. They, indeed, should clearly explain that an interaction of VDAC1 with mutated forms of SOD1 was found exclusively in a specific form of the inherited pathology (ALS type 1) and has been demonstrated for specific SOD1 mutations (i.e. G93A, G85R). In fact, mutations in SOD1 gene account for about 20% of the genetic ALS cases that, in turn, represent just 10% of the overall cases. Authors should also use the full name for SOD1 and not just the acronym.

We now modified the sentence as follows: “Conversely, a reduction in HKI concentration in the spinal cord is thought to enhance binding of VDAC1 to specific amyotrophic lateral sclerosis type I-associated variants of superoxide dismutase 1 (SOD1), thereby promoting formation of toxic SOD1 aggregates, mitochondrial dysfunction and cell death in motor neurons” (p. 2, lines 64-67).

3) In the sentences in line 74-76, the authors should take into account that the involvement of VDAC1 E73 residue in interaction with HK1 has been already proposed by Magri and coauthors. In the paper, already cited by the authors (Ref. no. 54), molecular dynamics simulations were carried out by using a small peptide mimicking the N-terminus of HK1.

We adapted the Introduction and now included a reference to the molecular docking simulation conducted by Magri et al. 2021 (Ref. 31), who provided evidence for a high-affinity binding site for a peptide mimicking the N-terminus of HKI on the outside wall of VDAC1 near the bilayer-facing E73 residue (p. 2, lines 77-79).

4) The sentence in line 109, “Combinatorial loss of VDAC1 and VDAC2 wiped out the mitochondria-associated pool of endogenous HKI”, referring to Fig. S2C, is not quite correct. Indeed, from Fig. S2D clearly emerges that the absence of VDAC1 and VDAC2 per se strongly affect the expression of endogenous HKs. Therefore, the subcellular fractionation must be contextualized with this result (i.e., the reduced mitochondrial pool of HKs is principally the result of a reduced expression of HKs in comparison to the other experimental conditions).

We now modified the text as follows: “Moreover, endogenous HKI protein levels were significantly reduced in VDAC1/2 double KO cells while subcellular fractionation experiments showed that in these cells, endogenous HKI primarily resides in the cytosol (Fig. S2c, d)”. See p. 3, lines 111-113.

5) To performing molecular dynamics simulations, authors added an additional Glu residue to the HKI N-terminal sequence of 17 residues long amino acid peptide, previously used in the

experiments. What is the rationale for this change?

The HKI structure has a straight helix up until residue 18, from where it kinks. The decision to use the full 18 residues in the MD-simulations (with residue 18 being a Gln, not a Glu) was a precaution to have sufficient C-terminal polar residues to both i) better represent any anchoring of the C- vs N-terminus to the membrane interface (in the full protein only the N-terminus can penetrate the membrane), and ii) preemptively satisfy peptide-water interactions, since it has been shown that short Martini 3 peptide helices may have a tendency towards excessive hydrophilicity in membrane contexts (Claveras Cabezudo et al., 2023 - Ref. 68; Spinti et al., 2023 - Ref. 69). This rationale has now been added to the system description (p. 15, lines 571-578).

6) Regarding the generation of the 3D model of VDAC2, the authors assumed that the structures of VDAC1 and VDAC2 are identical (as reported in Methods, line 529), despite several differences in the amino acid composition. Based on this consideration, they mutated the structure of VDAC1 by including the additional 11-amino acid portion of VDAC2 at the N-terminal domain. This, however, appears to me a questionable approach since a predicted 3D structure, such as that generated by the popular software AlphaFold, is available.

The description of the procedure we used to create a 3D model of VDAC2 for MD simulations appears to give rise to confusion. We believe the reviewer interpreted the text to mean that the first 11 residues of VDAC2 were grafted onto the VDAC1 structure. In reality, we ignored those 11 residues, and started mutating VDAC1's 1st residue as if it were VDAC2's 12th, and so forth. This effectively created a VDAC2 structure with the 11 N-terminal residues truncated off, which we also used in a previous study (Dadsena et al., 2019 - Ref. 10). We have now clarified the text in this regard.

Unfortunately, AlphaFold does not provide a reliable structural estimate of VDAC2's N-terminal region (residues 1-11), which has very low confidence scores and an unlikely fold. However, this segment is likely to remain in the channel's lumen and/or its aqueous vicinity, and thus of little consequence to the membrane-mediated interactions we observe. As stated in the Methods section, we found a C-alpha RMSD between human VDAC1 and zebrafish VDAC2 of 2.03Å. From this value, we conclude that the secondary structure of VDAC1 and our truncated VDAC2 are similar enough for using VDAC1 as a basis on which to mutate VDAC2 on.

7) The accession number reported for VDAC2 (line 526) associates with a protein long 310 residues, whereas is known that the main VDAC2 isoform is composed of 294 residues. Please correct.

We thank the reviewer for pointing out this mistake. As now indicated in the text, we used the sequence of human VDAC2 provided by UniProt entry P45880.

8) The authors should clearly explain the rationale behind the use of Leu7Gln mutation in CG-MD experiment (lines 152-154) to explain the changes of residence time along the membrane. This approach, in my opinion, is not immediately clear.

We now added the following explanation to the main text: "Leu7 is a key component of the membrane-oriented HKI-N apolar face, sitting at its very center (Fig. 2b), and thus likely in constant contact with the hydrophobic membrane core" (p. 4, lines 154-156).

9) It is supposed that the pH acidification should not affect the co-localization of blue and magenta signals. However, in Fig. 4C, at pH 6 the colocalization is not appreciable, as instead it is in the Fig. S5D.

While the relative intensity of the blue signal (OMM-Cherry) at sites where it colocalizes with the

magenta signal (Tom20-Halo) is somewhat dimmer at pH 6.0 and pH 7.4, the line scans in Fig. 4C show a degree of overlap between the two signals that is virtually indistinguishable from that in OPTIMEM. This indicates that the level of colocalization between OMM-Cherry and Tom20-Halo remains largely unaffected by fluctuations in pH.

Minor comments:

1) *In line 43, another reference should be added (PMID: 22020053).*

This reference has now been included.

2) *The explanation of Fig. 2C in line 142 is not clear. From the figure, it looks like that the apolar face takes contact with the polar head of phospholipids.*

We have modified Fig. 2C so that the apolar face of HKI-N is now penetrating deeper into the phospholipid leaflet.

3) *In line 74, please correct “significant” with “significan”.*

This has now been corrected.

Reviewer #3

The presented work aims to shed light into the mechanism of binding of hexokinase KHI to the mitochondrial voltage-dependent anion channels VDAC1/2. To achieve this goal the authors combine both computational simulations and experimental results. The authors provide evidence for several of the steps likely to be relevant for the binding, such as the importance of the presence of a bilayer-facing negatively charged Glutamic acid present in both VDAC1 and VDAC2, the importance of the N-terminal alpha helix of HKI and the orientation that these channels are most likely to take to allow the binding.

The authors provide an exhaustive and well-designed analysis of the different questions at hand combining coarse grained MD analysis with experimental validation for each step. However, the manuscript could benefit from more clear explanations of the steps followed and analysis performed, in order to improve interpretability and reproducibility of the presented results.

My comments would be structured as follows: minor format corrections such as typos and consistency, minor corrections and optional suggestions to improve the manuscript.

Format corrections:

1) *The way that mutations are written in the text is confusing. Usually, one writes first the amino acid that is presented in the wildtype followed by the mutated residue. The authors write those abbreviations correctly when using the one letter codes of the amino acids (for example VDAC1 E73F/F71E in line 226) but they write it the other way around when writing it in an extended way. For example, in line 223-224 “we substituted Phe for Glu73 and Glu for Phe71” should be “we substituted Glu73 for Phe and Phe71 for Glu”. This is true for all the mutations described in the manuscript.*

We believe the reviewer’s remark is related to two opposite use-cases of “substitute”. In the text we use “substitute B for A” to mean “A was replaced by B”. By contrast “substitute B by A”, which we do not use in the text, is usually taken to mean “B was replaced by A”. An internet search for the verbatim phrase “substituted Ala for” yields several examples with the same meaning as in our text. In any case, we will be glad to take editorial guidance in case this writing style should be avoided.

2) *There is a typo in the methods section at line 567, seiisotropically should be semiisotropically.*

This has now been corrected.

3) *There is no citation for HeliQuest on the manuscript.*

This has now been corrected.

4) *The references for each PDB structure should be added next to the PDB codes.*

This has now been corrected.

Minor corrections:

The authors performed a quite exhaustive set of analyses. However, in some sections it is difficult to follow what the authors did or how they arrived to those conclusions. Additionally, a more quantitative analysis of the results would help even if the final conclusion is qualitative. The following comments aim to improve on those aspects:

1) *The initial placement and restraining protocol of HKI-N for the adsorbition MD studies should be explained in more detail even if it was already described in citation 38. Similarly to how the authors summarize the Titrable Martini simulations setup described in ref 39. The initial placement and restrain setup can heavily influence the final results so it merits a more detail explanation.*

We have now dedicated one paragraph in the Methods section to describe the peptide restriction protocol in more detail (p. 16, lines 629-645).

2) *On line 152, where the authors study the importance of the apolar face of HKI-N in membrane binding, why was the residue Leu7 selected? All other mutations studied in the manuscript have a detailed explanation of why they were picked (previous knowledge, interactions observed ...). However, this mutation has no apparent reason for being selected. Were other positions studied? Is there any specific analysis performed to select this position? The authors should explain this in more detail.*

We now added the following explanation to the main text: “Leu7 is a key component of the membrane-oriented HKI-N apolar face, sitting at its very center (Fig. 2b), and is thus likely to be in constant contact with the hydrophobic membrane core” (p. 4, lines 154-156).

3) *For the desorption analysis, one should be careful when using only one replicate and using total residence times. It would be also beneficial if the authors added to the text the increments between the helix and membrane distance which were observed several times on the L7Q mutant before the complete desorption criteria was met (figure 2d).*

Please note that for the desorption analysis, a total of six replicates was used for each condition (WT, LQ7), with the stills in Fig. 2d corresponding to a representative image from one of the replicas of each condition. The plots in Fig. 2d show the 6 replicate traces plotted simultaneously; each of those apparent increments is one of the traces leaving the membrane, after which it is no longer drawn. Prior to membrane departure, the amplitude of membrane z-distance fluctuations is in the order of angstroms. To clarify the simultaneous plotting, we now adjusted the legend to Fig. 2d.

4) *On the paragraph starting at line 176, the authors should add numeric results (fractions of time) extracted from the MD simulations when talking about the frequency or stability of the contacts. It*

would also be beneficial to add a measure of uncertainty for example by doing the same analysis over different chunks of the obtained trajectories.

We appreciate the reviewer's comment on frequency and statistics. This led us to include the numeric results (fractions of time \pm SEM) as requested (p. 5, lines 184-190; p. 6, lines 196-201). We now also plotted the contact frequency and duration as shown in new Suppl. Fig. S4) and discuss our findings in the context of mutations to VDAC1's polar interface (p 8, lines 286-292).

5) On the "VDAC channels cause thinning of the lipid monolayer proximal to the membrane-buried Glu" section, more numerical results from figure 6 should also be added on the text. For example, the authors comment that they observed a membrane thinning reaching about 0.8 nm. In here it would be helpful to provide the membrane size in the conditions where thinning was not observed.

We thank the reviewer for pointing this out. We have now simulated a 100% POPC membrane without a VDAC inserted in it and calculated its average leaflet thickness. We also included minimum thickness values and statistics from the three replicates. Furthermore, we mapped the general membrane thinning capacity of VDACs outside of the region adjacent to the membrane-buried Glu in the cytosolic leaflet as shown in new Suppl. Fig. S8. We have now added references to the newly calculated values on p. 7, lines 244-257.

6) For the membrane thinning analysis, was the whole simulation used or were some frames discarded as equilibration? How uncertain are the estimated averages? What are the observed average membrane sizes over different chunks of the trajectory? This kind of information would support the reliability of the results.

We have used the complete production run after an equilibration of 2 ns. As these are 100% POPC membranes, we believe that they are quickly equilibrated and the production run can be immediately used for analysis. We have now added the SEM's of the minimal thickness value from each replica as an indicator of statistical uncertainty. We have done the same for the average membrane thickness (p. 7, lines 244-257).

7) For the "Polar residues proximal to the membrane-buried Glu provide a gateway for HKI-VDAC binding" section, on line 257 I would suggest adding that the described results come from simulations of the mutated Thr77 and Ser101 systems.

We now modified the text to indicate that the described results come from simulations (p. 7, lines 274-276).

8) Similar to my previous points, for the frequency, contacts and membrane thinning, the authors should add some numeric results to the text to facilitate comparison and provide statistical uncertainty of the obtained results. The uncertainty point is even more important in this section, where the effects observed are less drastic than the ones described on the other sections.

We have now added the minimum leaflet thickness value of the double-mutant and included SEM of the replicates as an indicator of how drastically the membrane thinning capability is reduced in the double-mutant compared to the wild-type (p. 8, lines 276-283). Additionally, we now plotted the changes in membrane thinning capacity of the single mutants and the double mutant compared to wild-type as a supplementary graph (Fig. S8 Panel C). We have also added numbers for HKI-N contact frequency in the double mutant, similarly as in comment #4 (p. 8, lines 286-292).

Optional comments:

1) It would be beneficial for the reproducibility and transparency if the authors could share the input

files (structures, scripts to set up the simulations and GROMACS input files such as topology files, mdp files and restrain specification files used on this work).

We have now created and uploaded an archive with all the requested files (doi: 10.5281/zenodo.14144611) and updated the Code Availability section in the manuscript.

2) Although in the supplementary information there is a list of all the simulations performed with their box sizes and simulation lengths, in my opinion it would be beneficial if in the methods section there is a summarized description of all the systems simulated. This would help illustrate all the work performed for the manuscript.

As per the reviewer's suggestion, we have added a cumulated description of all simulated systems in the MD simulation settings section of the Methods.

Response to reviewers' comments

Reviewer #1

[Original comment] p9, lines 343-345: *Is alkalinity a cause or effect in malignancy? How does stabilization of VDAC-HKI complex give rise to rapid growth of tumor cells? Is there any time frame beyond which the aforesaid complex is tumor causing? A brief explanation highlighting the mechanism involved for tumor growth in this case is desirable.*

As requested by the reviewer, we now modified the Discussion and added 3 new references to highlight how cellular alkalinity may promote rapid growth of tumor cells growth in the context of our findings (p10, lines 397-414, with textual changes marked in red):

Binding of HKI to mitochondrial VDACS has important physiological consequences, from modulating inflammatory responses to promoting cell growth and survival in highly glycolytic tumors. Multiple studies revealed that binding of HKI to VDACS protect tumor cells from permeabilization of the OMM and cytosolic release of cytochrome *c*, an event that marks a point of no return in mitochondrial apoptosis^{17,18,51}. **Mitochondria-bound HKI confers apoptosis resistance in human colon cancer cells by accelerating retrotranslocation of truncated BID, BAX and BAK from mitochondria⁵²**. Binding of HKI to mitochondrial VDACS also determines whether the product of the enzyme (G6P) is catabolized through glycolysis or shunted through the anabolic pentose phosphate pathway (PPP). While dissociation of VDAC-HKI complexes shifts the glucose flux to the PPP, leading to increased inflammation and decreased cell survival¹¹, mild alkalization of cytosolic pH pushes glucose metabolism toward glycolytic flux by augmenting VDAC-HKI binding⁴⁹. **Cellular alkalinity is a hallmark of malignancy⁵³. Increased intracellular pH is an early event in cancer development⁵⁴ and can induce dysplasia in the absence of activated oncogenes⁵⁵. Its stabilizing effect on VDAC-HKI complexes may facilitate disease progression by promoting glycolysis and apoptosis resistance, thus providing rapidly growing tumor cells with important metabolic and survival benefits.** In this context, the oncogenic potential of a somatic missense mutation p.E73D in VDAC1 identified in a colon adenocarcinoma (COSV54738458; cancer.sanger.ac.uk/cosmic) merits further investigation given our present finding that it promotes HKI binding.